# Dendritic cell MST1 inhibits Th17 differentiation

Chunxiao Li[1,2,*], Yujing Bi[3,*], Yan Li[1,2,*], Hui Yang[2,*], Qing Yu[1], Jian Wang[1,2], Yu Wang[1,2], Huilin Su[1,2], Anna Jia[1], Ying Hu[1], Linian Han[1], Jiangyuan Zhang[1], Simin Li[1], Wufan Tao[4] & Guangwei Liu[1,2]

Although the differentiation of CD4$^+$T cells is widely studied, the mechanisms of antigen-presenting cell-dependent T-cell modulation are unclear. Here, we investigate the role of dendritic cell (DC)-dependent T-cell differentiation in autoimmune and antifungal inflammation and find that mammalian sterile 20-like kinase 1 (MST1) signalling from DCs negatively regulates IL-17 producing-CD4$^+$T helper cell (Th17) differentiation. MST1 deficiency in DCs increases IL-17 production by CD4$^+$T cells, whereas ectopic MST1 expression in DCs inhibits it. Notably, MST1-mediated DC-dependent Th17 differentiation regulates experimental autoimmune encephalomyelitis and antifungal immunity. Mechanistically, MST1-deficient DCs promote IL-6 secretion and regulate the activation of IL-6 receptor $\alpha/\beta$ and STAT3 in CD4$^+$T cells in the course of inducing Th17 differentiation. Activation of the p38 MAPK signal is responsible for IL-6 production in MST1-deficient DCs. Thus, our results define the DC MST1–p38MAPK signalling pathway in directing Th17 differentiation.

[1] Key Laboratory of Cell Proliferation and Regulation Biology of Ministry of Education, Institute of Cell Biology, College of Life Sciences, Beijing Normal University, Beijing 100875, China. [2] Department of Immunology, School of Basic Medical Sciences, Fudan University, Shanghai 200032, China. [3] State Key Laboratory of Pathogen and Biosecurity, Beijing Institute of Microbiology and Epidemiology, Beijing 100071, China. [4] State Key Laboratory of Genetic Engineering and Institute of Developmental Biology and Molecular Medicine, Collaborative Innovation Center of Genetics and Development, School of Life Sciences, Fudan University, Shanghai 200433, China. * These authors contributed equally to this work. Correspondence and requests for materials should be addressed to W.T. (email: wufan_tao@fudan.edu.cn) or to G.L. (email: liugw@bnu.edu.cn).

CD4$^+$T cells are an essential component of the adaptive immune system and regulate immune responses to foreign antigens[1–6]. The activation and differentiation of CD4$^+$ T cells are regulated by the three main signalling components of the T-cell receptor (TCR) (signal 1), co-stimulatory molecules (signal 2) and cytokine receptors (signal 3)[4–7]. These signals depend on the regulatory role of innate immune cells. In the presence of cytokines produced by innate immune cells, naive CD4$^+$T cells differentiate into helper T-cell subsets with distinct functions and cytokine profiles. These include interferon-γ (IFNγ)-producing type 1 helper T (Th1) cells, which are essential for immunity to intracellular microorganisms, IL-4-producing Th2 cells, which protect against parasites and extracellular pathogens[4], and Th17 cells that produce IL-17A, IL-17F, IL-21 and IL-22 and protect against bacterial and fungal infections at mucosal surfaces[8].

Dendritic cells (DCs) are professional antigen-presenting cells (APC) that bridge innate and adaptive immunity. In addition to presenting antigens and modulating cell surface co-stimulatory molecules, DC-derived cytokines and chemokines can be proinflammatory or anti-inflammatory, and can engage distinct T-cell differentiation programs[9]. For example, the binding of the proinflammatory cytokine IL-6 to a complex of the IL-6 receptor α (IL-6Rα, also known as CD126) and IL-6Rβ (CD130; signal transducing receptor gp130) activates the transcription activator STAT3, resulting in differentiation of naive CD4$^+$T cells into Th17 cells by inducing the lineage-specific transcription factor RORγt[10–15]. Studies from our lab and others have shown that innate signalling in DCs mediated by G protein-coupled receptor S1P1 (refs 16,17), sirtuin 1 (ref. 18), mitogen-activated protein kinase (MAPKs)[19,20] and Wnt-β-catenin[21] has a critical role in shaping adaptive immune responses by directing naive CD4$^+$ T-cell differentiation. How the differentiation of CD4$^+$T cells is modulated and regulated by innate immune signals in DCs remains to be understood.

Mammalian sterile 20-like kinase 1 (MST1) is mammalian class II germinal center protein kinase, also known as serine/ threonine kinase 4 and kinase responsive to stress 2 (refs 22,23). MST1 has been implicated in regulating the cell cycle and apoptosis in various species[24–29]. MST1 is also involved in regulating adaptive immune cell function[30,31]. MST1-deficient mice accumulate mature lymphocytes in the thymus and have low numbers of naive T cells in the peripheral lymphoid organs due to a dysregulation of chemotaxis and apoptosis[32–34]. MST1 controls the development and function of regulatory T (Treg) cells through modulation of Foxo1/Foxo3 stability in autoimmune disease[35]. In addition, MST1 regulates the activation of T cells by phosphorylating the cell cycle inhibitory proteins MOBKL1A and MOBKL1B[36]. Furthermore, MST1 is important for optimal reactive oxygen species (ROS) production and bactericidal activity of phagocytes because it promotes the activation of the small GTPase Rac as well as mitochondrial trafficking and juxtaposition to the phagosome through the assembly of a TRAF6–ECSIT complex[37]. However, whether MST1 is involved in bridging the innate immune signal to the adaptive immune response is not clear.

Here, we show that MST1 has a critical role in directing the T-cell lineage fate by producing DC-derived cytokines, which link innate and adaptive immune modulation. Through a p38MAPK–MK2/MSK1–CREB dependent signalling pathway, MST1 is required for IL-6 production by DCs as well as for the expression of IL-6Rα/β and phosphorylation of STAT3 in responding T cells, resulting in specific lineage engagement of Th17 cells in experimental autoimmune encephalomyelitis (EAE) and fungal infection-induced inflammation.

## Results

**Deficiency of MST1 in DCs does not alter DC homoeostasis.** To investigate the role of MST1 in the immune system, we purified many types of mouse immune cells including macrophages (CD11b$^+$F4/80$^+$ cells), DCs (CD11c$^+$MHCII$^+$F4/80$^−$ Ly6G$^−$NK1.1$^−$CD19$^−$TCR$^−$ cells), neutrophils (CD11b$^+$ Ly6G$^+$ cells), CD4$^+$T cells (CD4$^+$TCR$^+$ cells) and CD8$^+$ T cells (CD8$^+$TCR$^+$ cells) as described previously[18] and analysed MST1 expression. This showed that MST1 is highly expressed in DC cells (Supplementary Fig. 1A). To study the role of MST1 in DCs, we generated CD11c$^+$ cell-specific MST1-deficient mice by crossing Mst1$^{flox/flox}$ with Cd11c-Cre (referred to as Mst1$^{\Delta DC}$ hereafter). As a result, a significant reduction of MST1 mRNA and protein was observed in DCs, but not in macrophages or neutrophils (Supplementary Fig. 1B,C), indicating that Mst1$^{\Delta DC}$ could be used for studies of DC function.

Next, we examined the composition of DC subsets in vivo. MST1 deletion in DCs showed a comparable percentage and CD11c$^+$ cell number, as well as CD11b$^+$CD11c$^+$, and CD8α$^+$ CD11c$^+$ cell subpopulations in the spleen compared with wild type (WT) (Supplementary Fig. 2A–C). MST1 deletion in DCs did not alter the percentages of other immune cells including macrophages, neutrophils, B cells and T cells (Supplementary Fig. 2D,E). Furthermore, the co-stimulatory molecule expressions displayed similar changes including those of MHCII, CD80, CD86, CD40 and CD54 in DC cells, even during LPS stimulation or on antigen processing (Supplementary Fig. 3). Moreover, DC MST1 deficiency does not affect the apoptosis or proliferation of DC cells (Supplementary Fig. 4). Altogether, these data show the deletion of MST1 in DCs does not alter DC homoeostasis.

**DC MST1-deficient mice exhibited altered Th17 cell responses.** While Mst1$^{\Delta DC}$ mice displayed no gross abnormalities within 6–8 weeks of birth, these mice started spontaneously dying 15 weeks after birth (Fig. 1a). In addition, Mst1$^{\Delta DC}$ mice displayed more weight loss (Fig. 1b) compared with WT control. Pathological analysis revealed a typical severely destructive inflammation in the liver and colon of Mst1$^{\Delta DC}$ mice (Fig. 1c). Finally, Mst1$^{\Delta DC}$ mice displayed a more pronounced autoimmune phenotype (CD4$^+$TCR$^+$CD44$^{high}$CD62L$^{low}$ cells) in the mesenteric lymph nodes (MLNs), spleen and peripheral lymph nodes (PLNs) (Fig. 1d and Supplementary Fig. 4C). Importantly, the deficiency of MST1 in DCs significantly promotes the percentage of Th17 but not Th1, Th2 and Foxp3$^+$CD4$^+$T cells (Treg cells) in MLNs, peyer's patch (PPs), intraepithelial lymphocytes (IELs) and lamina propria lymphocytes (LPLs) (Fig. 1e,f and Supplementary Fig. 5). Next, we created Mst1$^{\Delta DC}$→WT and WT→WT complete chimeras and Mst1$^{\Delta DC}$; WT mixed chimeras (1:1) to determine whether MST1 deficiency in DCs would lead to these alterations. Mst1$^{\Delta DC}$→WT complete chimera mice resulted in earlier death and weight loss, like the phenotype in the Mst1$^{\Delta DC}$ mice, but the Mst1$^{\Delta DC}$-WT mixed chimeras' mice were largely normal (Supplementary Fig. 6). Therefore, we concluded that MST1 deficiency in DCs did lead to altered Th17 cell responses and autoimmune phenotypes.

**DC MST1 deletion enhances Th17 differentiation in EAE.** We observed the effects of MST1 deficiency in DCs in EAE, a typical Th1 and Th17-dependent autoimmune disease model[38–41], to determine whether MST1 functions in innate DC cells to regulate adaptive immunity. After immunization with a myelin oligodendrocyte glycoprotein peptide comprised of amino acids 35–55 (called simply 'MOG' here), Mst1$^{\Delta DC}$ displayed much greater disease progression (Fig. 2a). Flow cytometry analysis showed that the central nervous system (CNS) of Mst1$^{\Delta DC}$ mice

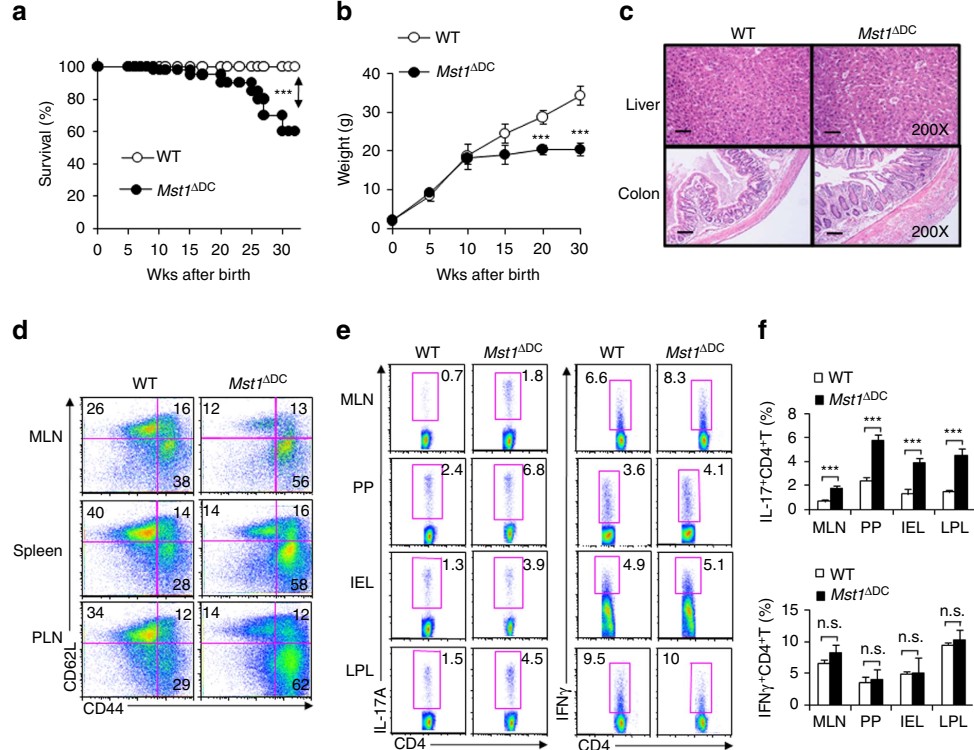

**Figure 1 | DC MST1 deficiency leads to spontaneous diseases.** (**a**) Kaplan–Meier plots of WT and $Mst1^{\Delta DC}$ mouse survival after birth are shown ($n = 20$). (**b**) Body weights of the WT and $Mst1^{\Delta DC}$ mice after birth are summarized ($n = 20$). (**c**) H&E staining of hepatic and intestinal tissues from sex- and age-matched WT and $Mst1^{\Delta DC}$ mice at 20 weeks after birth is presented. Scale bars, 100 μm. (**d**) DC MST1 deficiency leads to more CD44$^{high}$CD62L$^{low}$ cells among CD4$^+$T cells than WT control mice at 20 weeks after birth. Gating strategies for determining the percentage of CD4$^+$TCR$^+$CD44$^{high}$CD62L$^{low}$ cells are showed in Supplementary Fig. 4C. Intracellular IL-17A and IFN-γ expression of CD4$^+$T cells isolated from MLN, PP, IEL and LPL in WT and $Mst1^{\Delta DC}$ mice with a typical figure displayed (**e**) and the frequencies of positive cells summarized (**f**). Data are representative of three to four independent experiments (mean ± s.d.; $n = 4$). \*\*\*$P < 0.001$, compared with the indicated groups. n.s., not significant. $P$-values were determined using Student's $t$-tests.

had significantly more inflammatory cell infiltration into the spinal cord, including CD4$^+$T cells and CD11b$^+$ myeloid cells (Supplementary Fig. 7). Thus, MST1 signalling in DCs was required for protection against autoimmune diseases of the CNS.

Moreover, CNS-infiltrating T cells from $Mst1^{\Delta DC}$ mice had more IL-17$^+$ cells, but normal levels of IFNγ$^+$ cells, IL-4$^+$ cells and Foxp3$^+$ cells among the CNS-infiltrating CD4$^+$T cells (Fig. 2b,c). Consistent with this, MST1 deficiency in DC resulted in significantly higher and sustained expression of IL-17A, IL-17F, IL-6Rα, IL-6Rβ (gp130), but the expression of IFNγ, IL-4 and Foxp3 were similar in the CNS-infiltrating T cells compared with WT DC control (Fig. 2d).

To determine whether an alteration in Th17 differentiation is responsible for the more severe EAE in the $Mst1^{\Delta DC}$ mice, we observed the T-cell responses in the draining lymph node (dLN) at the preclinical stage of EAE on day 7 after immunization. MST1 deficiency in DCs does not alter T-cell proliferation (Fig. 2e and Supplementary Fig. 8), but significantly enhances the IL-17$^+$ cell percentage, IL-17 mRNA expression and IL-17 secretion compared with WT mice, whereas the IFNγ$^+$ cell percentage and expression are similar in the two group mice immunized with MOG (Fig. 2e–h). These data suggest MST1 deficiency plays a critical role in promoting Th17 differentiation *in vivo*, thus contributing to EAE.

**DC MST1 precipitates antifungal Th17 cell responses.** Next we challenged $Mst1^{\Delta DC}$ mice with *Candida albicans*, a fungal infection model known to induce a strong Th17 cell response. Nine days after infection, $Mst1^{\Delta DC}$ mice displayed significantly attenuated

pathological kidney injuries, less fungal survival in the kidney and higher serum IL-17 production, but not IFNγ production, indicating that MST1 signalling is required for precipitating the fungal infection in DCs (Fig. 3a–c).

Whether the Th17 response is related to antifungal immunity in $Mst1^{\Delta DC}$ mice, we investigated the T-cell response of splenocytes on day 9 after infection. MST1 deficiency in the DC did not alter the T-cell proliferation (Fig. 3d,e), but did significantly enhance the IL-17$^+$ cell percentage compared with the WT mice, whereas the percentages of IFNγ$^+$ cells, IL-4$^+$ cells and Foxp3$^+$ cells were similar in the two groups of infected mice. Consistent with this, the mRNA expression of IL-17A, IL-17F, IL-6Rα/β, but not IFNγ, IL-4 or Foxp3 were significantly enhanced (Fig. 3f). Thus, these data collectively suggest MST1 deficiency in DCs results in enhanced Th17 cell responses against fungal infection.

**DC MST1 directs antigen-specific Th17 differentiation.** We first observed the endogenous CD4$^+$T cells response following antigen immunization to investigate how DC MST1 directs antigen-specific T-cell differentiation. The T cells from $Mst1^{\Delta DC}$ mice produce higher amounts of IL-17, but not IFNγ (Fig. 4a). Furthermore, we used adoptive transfer experiments to observe the T cells responses induced by MST1-deficient DCs. We transferred naive CD4$^+$T cells (CD44$^{low}$CD62L$^{high}$; Thy1.1) from 2D2 mice (TCR specific for MOG peptide) into WT and $Mst1^{\Delta DC}$ mice, and then immunized the mice with MOG in complete Freund's adjuvant (CFA). The donor cells were analysed on day 9 after immunization. Donor T cells from the $Mst1^{\Delta DC}$

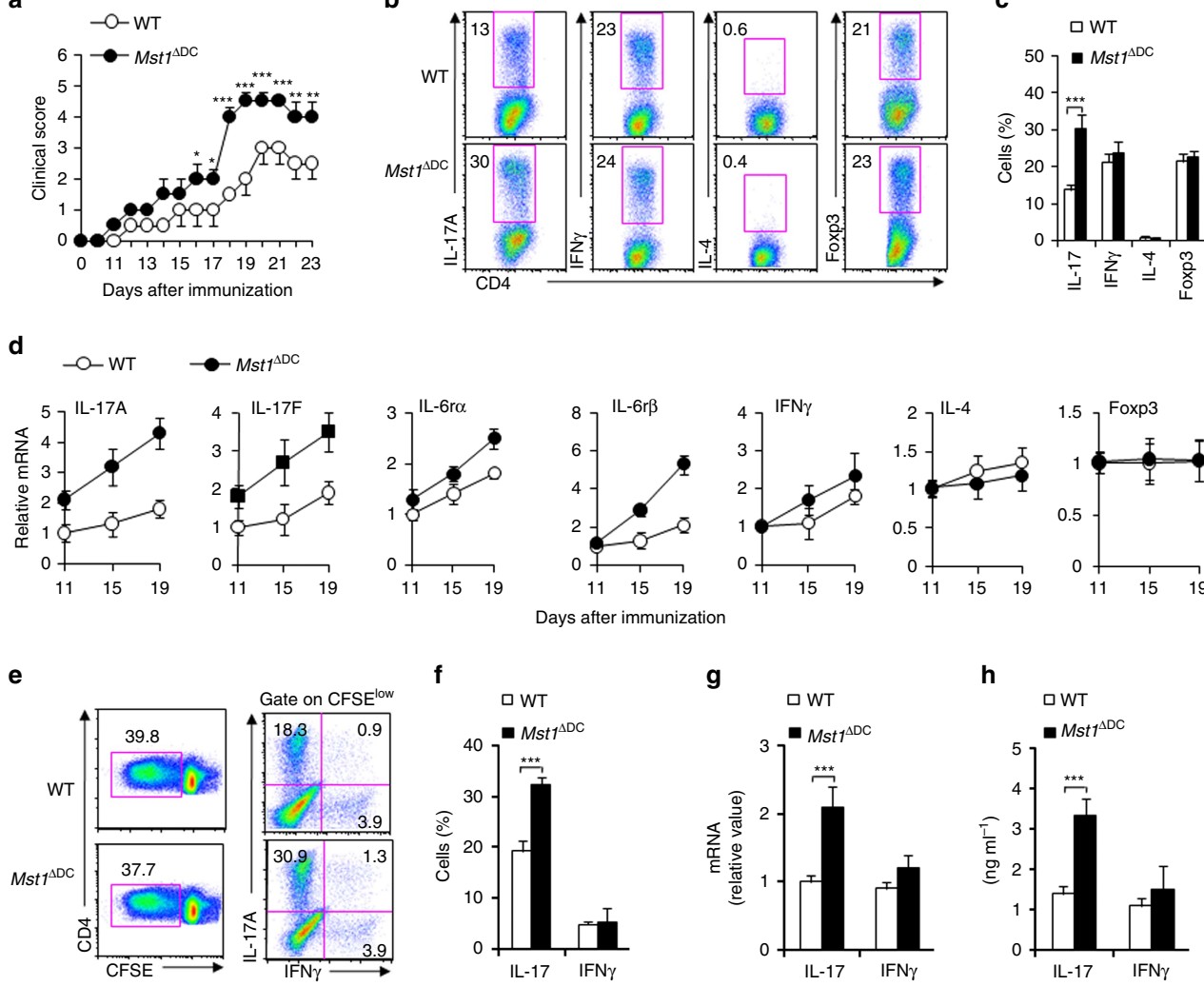

**Figure 2 | DC MST1 inhibits Th17 differentiation in EAE.** (**a**) EAE disease course in WT and $Mst1^{\Delta DC}$ mice. Intercellular staining of IL-17$^+$ cells, IFNγ$^+$
cells, IL-4$^+$ cells (after stimulation with PMA and ionomycin) and Foxp3$^+$ cells among CD4$^+$ T cells from the spinal cord of WT or $Mst1^{\Delta DC}$ mice
were determined with flow cytometry method (FCM) on day 19 after MOG immunization. A typical figure is shown (**b**) and the data summarized (**c**).
(**d**) CD4$^+$ T cells isolated from spinal cord of $Mst1^{\Delta DC}$ mice on the indicated day after MOG immunization along with mRNA expression of the indicated
gene (levels in WT groups were set to 1). (**e,f**) CD4$^+$ T cells isolated from dLN of MOG-immunized WT or $Mst1^{\Delta DC}$ mice were labelled with CFSE and
stimulated with MOG for 5 days. Intercellular staining of IL-17 and IFNγ in CFSE$^{low}$ cells was determined with FCM. The typical figure is shown in **e** and data
summarized (**f**). (**g,h**) CD4$^+$ T cells isolated from the dLN of MOG-immunized WT or $Mst1^{\Delta DC}$ mice, and ex vivo stimulation with anti-CD3 (1 μg ml$^{-1}$) for
24 h and mRNA expression (**g**) or cytokine secretion (**h**) of the indicated gene were analysed using qPCR (levels in the WT groups were set to 1) or ELISA.
Data are representative of three to four independent experiments (mean ± s.d.; $n = 4$-6). *$P < 0.05$ and ***$P < 0.001$, compared with the indicated groups.
$P$-values were determined using Student's $t$-tests.

recipients exhibited a similar proliferation level compared with
the WT recipients (Fig. 4b). However, donor T cells from
$Mst1^{\Delta DC}$ recipients displayed more IL-17$^+$ cells, IL-17 secretion
and mRNA expression compared with the WT recipients
(Fig. 4c,d and Supplementary Fig. 9). The percentages of IFNγ$^+$
cells and IFNγ levels were similar in the two groups.

Next, we extended MOG antigen-specific experiments to other
antigens to observe the T-cell responses. We used the ovalbumin
amino acids 323–339 (called simply 'OVA' here) antigen to
perform this experiment. Naive CD4$^+$ T cells (as described
above) from OT-II mice (TCR specific for OVA) had been
previously transferred into the WT and $Mst1^{\Delta DC}$ recipient mice
immunized with OVA in CFA. The donor cells were analysed on
day 9 after immunization. Donor T cells from the $Mst1^{\Delta DC}$
recipients exhibited similar proliferation activity compared
with that of the WT recipients (Fig. 4e). However, donor OT-II

T cells from $Mst1^{\Delta DC}$ recipients displayed enhanced Th17
differentiation compared with the WT-recipient mice (Fig. 4f,g).
Collectively, these results suggest that MST1 deficiency in DCs
promotes antigen-specific Th17 cell differentiation in vivo.

**MST1 is required for DC-dependent Th17 differentiation.**
Next, we applied an in vitro coculture system to investigate the
role of splenic DC to antigen-specific T cells, including purified
antigen specific T cells and splenic DCs and the stimuli LPS and
antigen. 2D2T cells and/or OT-II T cells stimulated by $Mst1^{\Delta DC}$
DCs in the presence of antigen displayed similar proliferation
activity compared with that of the WT DC (Supplementary
Fig. 10A,B). The expression levels of IL-2, CD25 and CD69 were
also similar in T cells co-cultured with the two kinds of DCs
in vitro, suggesting the MST1-deficient in DC doesn't alter the
T-cell activities (Supplementary Fig. 10C). However, we observed a

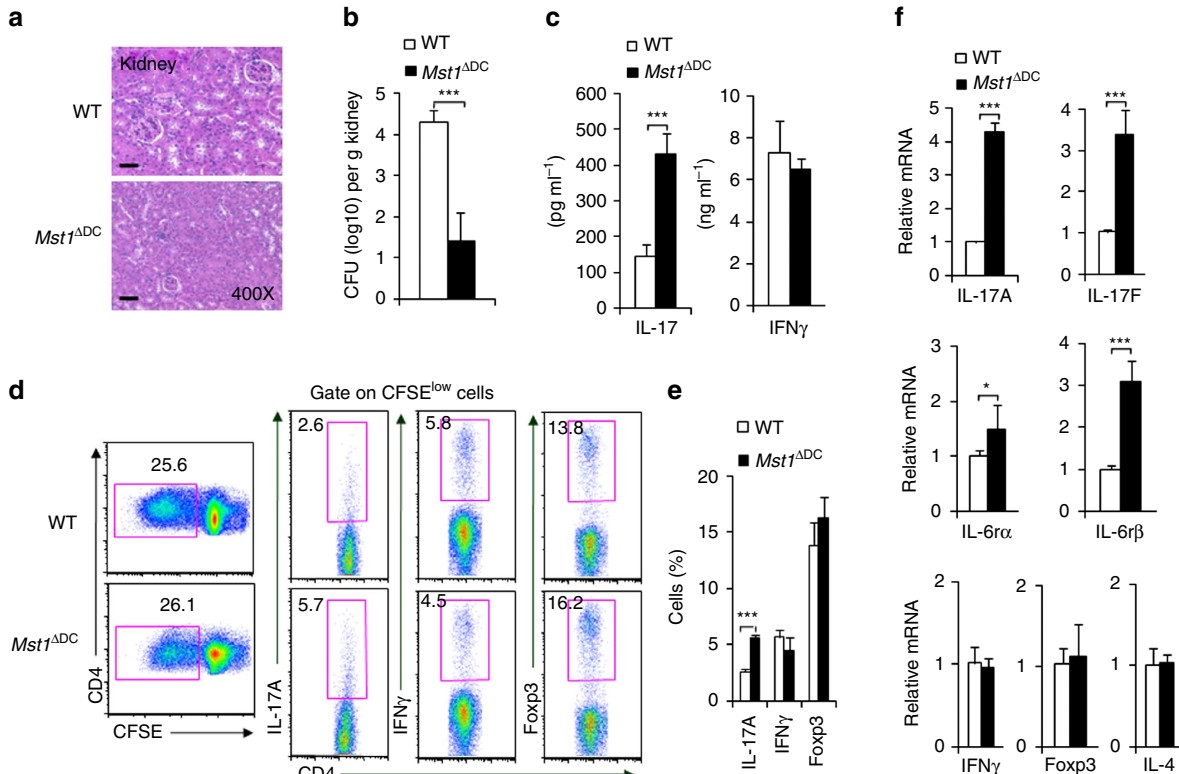

**Figure 3 | DC MST1 inhibits Th17 differentiation in anti-fungi infection.** Mice were infected with $10^5$ *C. albicans* SC5314 by i.v. injection. After 9 days, kidneys were collected and a photo of the H&E staining of pathological kidney injuries is shown (Scale bars, 100 μm) (**a**) and the fungal burden in the kidneys expressed as CFU per g (**b**). (**c**) Serum cytokine level of IL-17 and IFNγ in infected mice. (**d,e**) CD4⁺ T cells isolated from the spleen of the WT or *Mst1*^ΔDC mice were labelled with CFSE and stimulated with heat-killed *C. albicans* for 3 days. Intracellular staining of IL-17 and IFNγ in CFSE^low cells was determined with FCM. A typical figure is shown (**d**) and the data summarized (**e**). (**f**) CD4⁺ T cells isolated from the spleen of infected WT or *Mst1*^ΔDC mice, and *ex vivo* stimulation was performed with anti-CD3 (1 μg ml⁻¹) for 24 h, then mRNA expression of the indicated gene was analysed using qPCR (Levels in the WT groups were set to 1). Data are representative of three to four independent experiments (mean ± s.d.; *n* = 4–8). *P < 0.05 and ***P < 0.001, compared with the indicated groups. P-values were determined using Student's *t*-tests.

higher level of IL-17⁺ and mRNA expression in OT-II T cells co-cultured with *Mst1*^ΔDC splenic DC, even with a variety of *Mst1*^ΔDC DC subsets (CD11b⁺ DCs and CD8⁺ DCs), than in T cells co-cultured with WT DCs (Fig. 5a and Supplementary Fig. 10D,E). Furthermore, polyclonal T cells activated by the anti-CD3 and MST1-deficient DC subsets displayed similar alterations in Th17 and Th1 differentiation (Fig. 5b). Moreover, the Th17-assoaciated cytokine IL-17A/F, cytokine receptor IL-6Rα/β and gene encoding the transcriptional factor RORγt (Rorc) all displayed higher and more sustained upregulation in the T cells co-cultured with *Mst1*^ΔDC DCs co-cultured with WT DCs (Fig. 5c). The other T-cell subtype-associated cytokines and transcriptional factors were found to be similar in the two kinds of T cells co-cultured with DCs (Fig. 5c). These data suggest that MST1 is required in DCs for the induction of Th17 cell differentiation.

To investigate whether DC MST1 is sufficient for deciding the fate of Th17 cells, we ectopically expressed MST1 with a retrovirus and sorted the positive green fluorescent protein (GFP) DCs. We co-cultured them with OT-II cells and observed the T-cell response as described above. Modest MST1 over-expression (MST1-RV) resulted in a lower IL-17⁺ percentage of T cells and IL-17 mRNA expression, but not IFNγ⁺ percentage or IFNγ mRNA expression compared with the control group (RV; Supplementary Fig. 11). These data suggest DC MST1 signalling is sufficient for inducing Th17 cell differentiation. Taken together, these data collectively suggest MST1 signalling in DCs, acting as a negatively regulator, directly inhibits Th17 cell differentiation.

**DC MST1 directs Th17 differentiation through IL-6.** Next, we analysed the DC-derived cytokine levels, including TNF, IL-4, IL-6, IL-10, IL-12, IL-18, IL-23, IL-1β, TGFβ and IFNα. MST1 deficiency resulted in significant higher IL-6⁺ cells among the DCs compared with WT DCs after LPS stimulation (Fig. 6a). Also, similar alterations of IL-6 secretion and mRNA expression were observed in the DCs after LPS stimulation. However, the secretion and mRNA expression of other cytokines were comparable in the *Mst1*^ΔDC and WT DCs after LPS stimulation *in vitro* (Fig. 6b and Supplementary Fig. 12). In contrast, modest expression of MST1 in DCs resulted in significantly lower IL-6 production compared with control, but the IL-23, IL-1β and TGFβ1 were similar in the two groups of DCs (Supplementary Fig. 13). Together, these data suggest IL-6 is likely to be critical for the Th17 cell differentiation induced by *Mst1*^ΔDC DCs.

We applied a DC-T-cell co-culture system to determine whether MST1 signalling in DCs regulates Th17 differentiation through IL-6. In this system, we observed that *Mst1*^ΔDCIL-6⁻/⁻ DCs almost completely reversed the impact of *Mst1*^ΔDC on Th17 differentiation (Fig. 6c). Consistently, we also were able to reverse the impact of MST1-RV on Th17 cell differentiation in our co-culture system by adding IL-6 (Fig. 6d). Altogether, these data suggest that IL-6 production in *Mst1*^ΔDC DCs as well as MST1-RV DCs contributes to Th17 cell differentiation.

To investigate the wide availability of DC-directed T-cell responses, we isolated DCs from the spinal cord and intestine and found that MST1 deficiency does not alter the composition of DCs, it dose significantly promote the secretion of IL-6

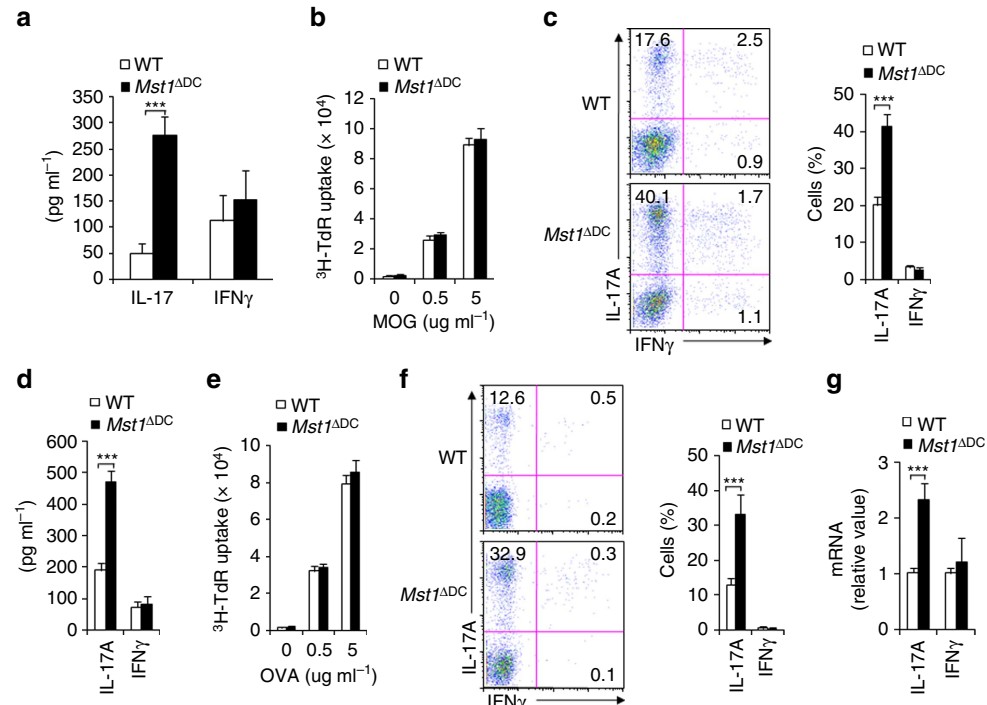

**Figure 4 | DC MST1 signalling directs Th17 differentiation *in vivo*.** (**a**) WT or *Mst1*$^{\Delta DC}$ mice were immunized with OVA + CFA for 7–8 days. The T cells were isolated from the dLN and restimulated *ex vivo* for 3 days with OVA (5 µg ml$^{-1}$) in the presence of irradiated splenocytes, and the secretion of indicated cytokines were analysed by ELISA. (**b–d**) Naive MOG-transgenic 2D2 T cells (Thy1.1$^+$) were transferred into WT or *Mst1*$^{\Delta DC}$ mice and immunized with MOG + CFA for 7–8 days. (**b**) Proliferation of dLN cells was determined with $^3$H-TdR incorporation. (**c**) Expression levels of IL-17A and IFNγ in donor cells in dLN. Right, proportion of IL-17A and IFNγ in donor cells. (**d**) Secretion of IL-17 and IFNγ in donor cells stimulated with MOG for 72 h. (**e,f**) Naive OT-II T cells (Thy1.1$^+$) were transferred into WT or *Mst1*$^{\Delta DC}$ mice and immunized with OVA + CFA for 7–8 days. (**e**) Proliferation of dLN cells was determined with $^3$H-TdR incorporation. (**f**) Expressions of IL-17A and IFNγ in donor cells in the dLN. Right, proportion of IL-17A and IFNγ in donor cells. (**g**) mRNA expression of IL-17 and IFNγ in donor cells stimulated with OVA for 72 h. Levels in WT control groups were set to 1. Data are representative of three to four independent experiments (mean ± s.d.; *n* = 3–6). ****P* < 0.001, compared with the indicated groups. *P*-values were determined using Student's *t*-tests.

and direct Th17 cell differentiation (Supplementary Figs 14 and 15).

**DC MST1 controls the signalling of IL-6R/p-STAT3 of T cells.** T-cell polarizing cytokines often induce the expression of their corresponding cytokine receptor on T cells, resulting in robust programing of cell fate determination[1,4]. This led us to investigate the expression of the cytokine receptor of IL-6 in T cells activated by WT and *Mst1*$^{\Delta DC}$ DCs. We applied a DC-T-cell co-culture system (as described above) to investigate this. *Mst1*$^{\Delta DC}$ DCs induced significantly higher expression of IL-6Rα and IL-6Rβ in T cells compared with WT DCs (Fig. 7a). The intercellular cytokine signalling in T cells also exhibited significant alteration. *Mst1*$^{\Delta DC}$ DCs induced higher phosphorylation of STAT3 compared with WT DCs in a DC-T-cell co-culture system. However, the phosphorylation of STAT4 was similar in the T cells co-cultured with *Mst1*$^{\Delta DC}$ DCs and WT DCs (Fig. 7b). These data suggest IL-6R-p-STAT3 signalling is probably involved in the modulation of Th17 cell differentiation that is induced by DC MST1.

To examine whether IL-6R signliang is necessary for the T-cell differentiation induced by DC MST1, we applied an siRNA approach to knockdown of IL-6Rα and IL-6Rβ expression in OT-II T cells (Supplementary Fig. 16A,C), which were subsequently co-cultured with WT or *Mst1*$^{\Delta DC}$ DCs in the DC-T-cell co-culture system as described above. Knockdown of IL-6Rα or IL-6Rβ consistently reversed the enhanced Th17 cell differentiation and phosphorylation of STAT3 by *Mst1*$^{\Delta DC}$ DCs, but Th1

differentiation was unchanged (Fig. 7c,d and Supplementary Fig. 16B,D). Thus, IL-6R-p-STAT3 signalling in T cells is required for the Th17 cell differentiation induced by DC MST1.

**DC MST1 controls IL-6 production through p38MAPK.** How does MST1 control cytokine production in DCs in directing T-cell differentiation? We assessed the activation of LPS downstream pathways, including ErK, c-jun-NH2-kinase (JNK), NF-kB, p38MAPK and AKT-mTOR by flow cytometry or immunoblot analysis to elucidate the mechanisms that mediate MST1 function in splenic DCs stimulated by LPS. As expected, LPS activated all the pathways in the WT DCs. In contrast, *Mst1*$^{\Delta DC}$ DCs activated the ErK, JNK, NF-kB and AKT-mTOR pathways like the WT DCs did, but a stronger and more sustained phosphorylation of p38MAPK (Fig. 8a,b and Supplementary Fig. 17). Moreover, modest MST1 overexpression (MST1-RV) resulted in lower phosphorylation of p38MAPK than control (RV; Supplementary Fig. 18). Thus, MST1 is associated with p38MAPK.

We pretreated WT and *Mst1*$^{\Delta DC}$ DCs with U0126 (an inhibitor of ErK) and SB203580 (an inhibitor of p38MAPK), respectively, followed by LPS stimulation. SB203580, but not U0126 significantly reversed the IL-6 production and IL-6 mRNA expression (Supplementary Fig. 19A,B) by blocking p38MAPK as indicated (Supplementary Fig. 19C). Furthermore, we used *Mst1*$^{\Delta DC}$p38$^{+/-}$ double knockout mice in this investigation (Supplementary Fig. 19D). Interestingly, the increased IL-6 expression and production in *Mst1*$^{\Delta DC}$ DCs were significantly

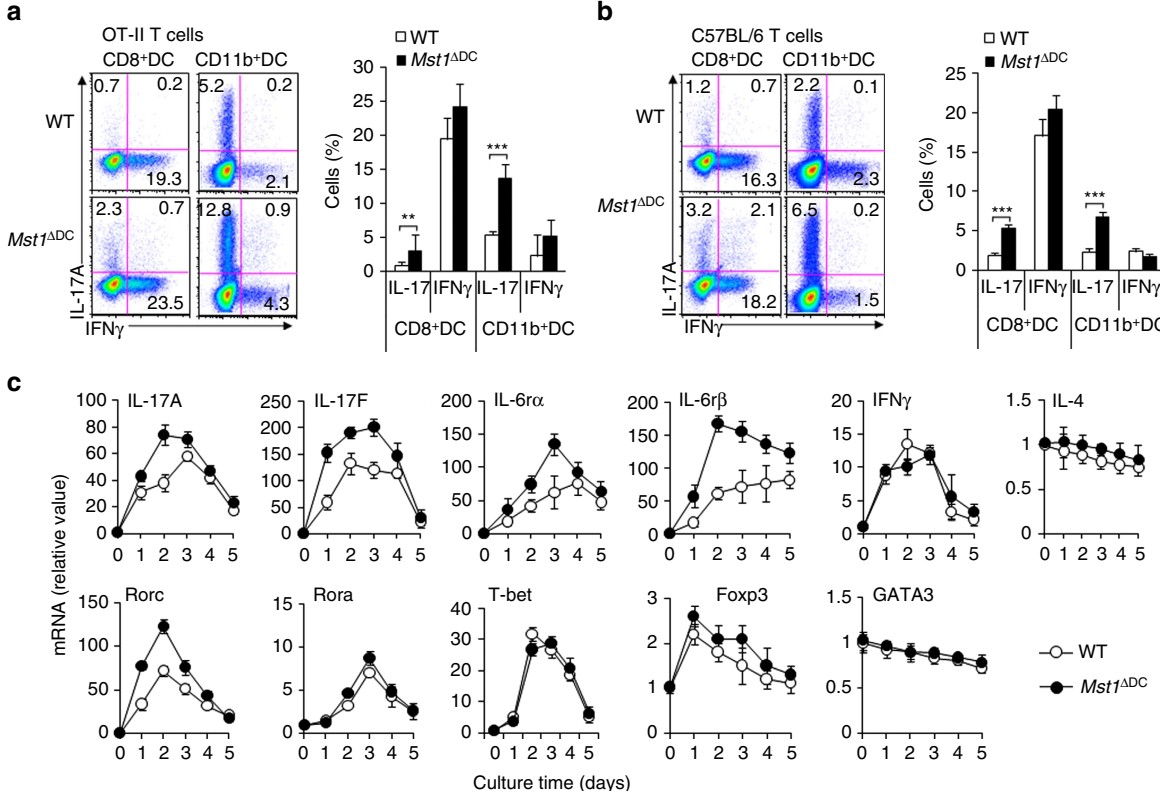

**Figure 5 | DC MST1 signalling instructs Th17 differentiation *in vitro*. (a)** Naive CD4+ T cells sorted from OT-II mice stimulated with antigen and LPS-pulsed CD8+DCs or CD11b+DCs from WT or *Mst1*ᐞDC mice for 5 days. Intercellular staining of IL-17 and IFNγ in T cells. Right, proportion of IL-17 and IFNγ in T cells. **(b)** Naive CD4+ T cells sorted from C57BL/6 mice stimulated with anti-CD3 (2 μg ml⁻¹) and LPS-pulsed CD8+DCs or CD11b+DCs from WT or *Mst1*ᐞDC mice for 5 days. Intercellular staining of IL-17 and IFNγ in T cells was determined with FCM and a typical figure is shown. Right, proportion of IL-17 and IFNγ in T cells. **(c)** Naive T cells from C57BL/6 mice were co-cultured with WT or *Mst1*ᐞDC splenic DCs for 5 days in the presence of anti-CD3 and LPS. mRNA expression of the indicated genes in T cells were determined with qPCR. Levels in the WT control groups were set to 1. Data are representative of three to four independent experiments (mean ± s.d.; $n = 4$–6). **$P < 0.01$ and ***$P < 0.001$, compared with the indicated groups. $P$-values were determined using Student's $t$-tests.

reversed in *Mst1*ᐞDCp38+/− double knockout DCs (Fig. 8c,d). The enhanced Th17 differentiation as well as IL-6Rα and IL-6Rβ expression observed in T cells co-cultured with *Mst1*ᐞDC DCs were also significantly reversed in T cells co-cultured with *Mst1*ᐞDCp38+/− double knockout DCs (Fig. 8e,f). Therefore, these data suggest the p38MAPK pathway is responsible for the IL-6 production during DC MST1-dependent Th17 cell differentiation.

We determined the activation of MAPK-activated protein kinase 2 (MK2) and mitogen and stress-activated protein kinase 1 (MSK1), two p38 targets[42,43] to investigate the signalling and transcriptioanl pathways involved in IL-6 production. Significantly unregulation of *Mst1*ᐞDC DC cell activation was observed compared with WT control cells (Supplementary Fig. 20A). Furthermore, cAMP response-element-binding protein (CREB)[44], which is important for IL-6 expression, was consistently upregulated in *Mst1*ᐞDC cells, whereas the phosphorylation of transcription factor C/EBPβ, c-Fos and IkBα unaltered (Supplementary Fig. 20A). Consistent with this, modest overexpression of MST1 in DCs resulted in lower and less inactivation of the phosphorylation of MK2, MSK1 and CREB (Supplementary Fig. 20B). Thus, MST1 is assoicated with the p38MAPK–MK2/MSK1–CREB signalling axis. Importantly, MK2/MSK1/CREB siRNA efficiently silenced their expression in DC cells and consistently reversed the IL-6 production and Th17 cell differentiation in *Mst1*ᐞDC DCs to a large extent (Supplementary Fig. 20C–K). These data collectively a

p38–MK2/MSK1–CREB signalling axis for the regulation of IL-6 production in DCs for Th17 cell differentiation.

**Knockdown of MST1 in human DCs directs Th17 differentiation.** Next, we sought to knockdown of MST1 with an siRNA approach targeting MST1 in both mouse and human DCs to see if this would recapitulate our findings in the genetic targeting MST1. MST1 siRNA efficiently knocked down the expression of MST1 in mouse DC cells, promoted the IL-6 secretion of DCs and upregulated the percentage of IL-17+ cells and the level of IL-6R expression in T cells with a DC-T coculture system (as described above). Moreover, blocking the p38MAPK pathway almost completely reverse the change (Fig. 9a–d). The siRNA experiment was then extended to a human DC-T-cell coculture system in which the DCs were derived from human peripheral blood monocytes and the T cells were isolated from human cord blood. Knockdown of MST1 in human DCs largely recapitulated the alteration in mouse DCs, including IL-6 production in DCs and IL-17A+ and IFNγ+ cell percentage as well as IL-6R expression in T cells (Fig. 9e–h). Thus, the data show that indicated MST1 mediates an evolutionarily conserved signalling pathway in both mouse and human DCs.

**Discussion**

A genetic approach was employed to selectively delete MST1 in DC cells and results showed that loss of MST1 in DCs causes

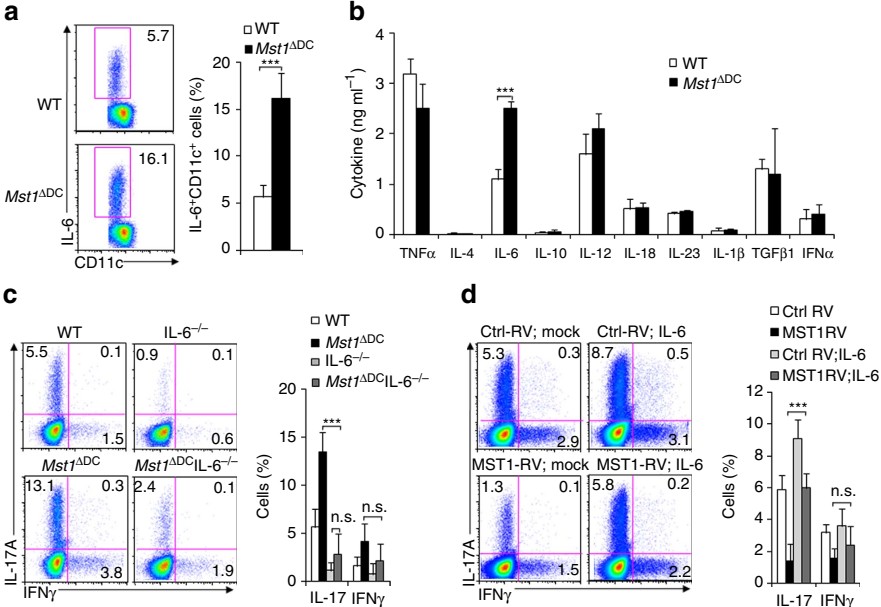

**Figure 6 | MST1 controls DC IL-6 production during Th17 differentiation.** (**a**) Intracellular staining of IL-6 in WT or *Mst1*<sup>ΔDC</sup> splenic DCs 5 h after LPS stimulation *in vitro*. (Right) Proportion of IL-6 among the CD11c⁺ cells. The supernatant was collected and cytokine secretions determined with ELISA (**b**). (**c**,**d**) CD4⁺T cells from the indicated mice stimulated with WT or *Mst1*<sup>ΔDC</sup> splenic DCs for 5 days. Intercellular staining of IL-17 and IFNγ in T cells was determined with FCM and a typical figure is shown. Right, proportion of IL-17 and IFNγ in T cells. Data are representative of three to four independent experiments (mean ± s.d.; *n* = 4–5). ***P < 0.001, compared with the indicated groups. n.s., not significant. *P*-values were determined using Student's *t*-tests.

massive inflammation in multiple tissues in old-aged mice and promotes Th17 cell differentiation in autoimmune or fungal infection inflammation. Conditional deletion of MST1 in mouse DCs or the use of siRNA in mouse or human DCs increased the IL-17 production from CD4⁺T cells, whereas ectopic MST1 expression in DCs inhibited IL-17 production of CD4⁺T cells. MST1-deficient DCs produced excess amount of IL-6, upregulated IL-6Rα/β expression and phosphorylated STAT3 in responding CD4⁺T cells and contributed to Th17 differentiation. In addition, we showed that MST1-deficient DCs exhibited an increased phosphorylation of p38MAPK–MK2/MSK1–CREB, which is responsible for IL-6 production in MST1-deficient DCs. Our study thus indicates MST1–p38MAPK–MK2/MSK1–CREB to be the critical signalling pathway in DCs for controlling IL-6 production and IL-6R-STAT3 expression in CD4⁺T cells in directing Th17 cell differentiation (Supplementary Fig. 21).

The kinase MST1 is the mammalian homologue of the *Drosophila melanogaster* kinase Hippo, which inhibits cell proliferation and promotes apoptosis during development[45–59]. MST1 deficiency in humans results in a complex, combined immunodeficiency syndrome with recurrent bacterial and viral infections, lymphopenia and variable neutropenia[60,61]. In mice, MST1 is an important regulator of the adhesion, migration, proliferation and apoptosis of cells[31–33]. In adaptive immune cells, MST1 is critical for T-cell migration, activation and function[34,35]. In innate immune cells, MST1 is important for the optimal ROS production and bactericidal activity of phagocytes by promoting activation of the small GTPase Rac and also mitochondrial trafficking and juxtaposition to the phagosome through the assembly of a TRAF6–ECSIT complex[37]. However, the interplay between adaptive immune cells and innate immune cells remains poorly understood. The present study shows that an integrated MST1–p38MAPK–MK2/MSK1–CREB signalling axis in DCs directs the generation of the Th17 subset in both autoimmune and fungal infection inflammation. Whereas MST1 is not involved in regulating antigen presentation of DCs,

the MST1–p38MAPK–MK2/MSK1–CREB axis in DCs instructs Th17 cell differentiation through its modulation of the DC-derived T-cell polarizing cytokine IL-6. The altered IL-6Rα/IL-6Rβ expression and downstream STAT3 signalling in responding T cells further confer a robust DC-T-cell cross-talk, directing the programming of specific Th17 cell differentiation.

DC-derived signals control adaptive immunity at multiple checkpoints that dictate the activation and differentiation of T-cell-mediated immune responses, but very few innate immune pathways have been shown to regulate Th17 responses[62,63]. Although some pathways have shown to be involved in the modulation of T cells, such as dectin-Syk-CARD9 signalling[64], MKP-1 signalling[19], Wnt-β-catenin signalling[21] and sirtuin1-HIF1α metabolism signalling[18], the essential innate cell-dependent Th17 modulation pathway still remains unclear. Previous studies have shown that p38 acts in a T-cell-intrinsic manner for Th17 differentiation, as IL-17 expression is diminished after pharmacological inhibition of p38 or dominant-negative p38 over-expression[65,66]. Other studies also showed that p38 is not responsible for T-cell-intrinsic modulation of Th17 differentiation[20]. It is reported that there is a selective role for p38 in DC-mediated Th17 differentiation but not in T-cell-intrinsic Th17 differentiation[20,67]. In accord with the latter studies, we found that DC MST1 deficiency significantly upregulated the activation of the p38MAPK–MK2/MSK1–CREB signal axis and contributed to the excess amount secretion of the T-cell polarizing cytokine IL-6 in directing Th17 cell differentiation.

It is intriguing that the germ line knockout of MST1 mice results in normal development, while the DC-specific MST1-deficient mice die as early as 15 weeks after birth. Similarly, the bone marrow chimeras (*Mst1*<sup>ΔDC</sup>→WT) mice also display an early death and weight loss, confirming the cell specificity of MST1 deficiency. Additional pathological and immunological analyses have revealed that the early death and weight loss in DC-specific MST1-deficient mice were due to a heightened

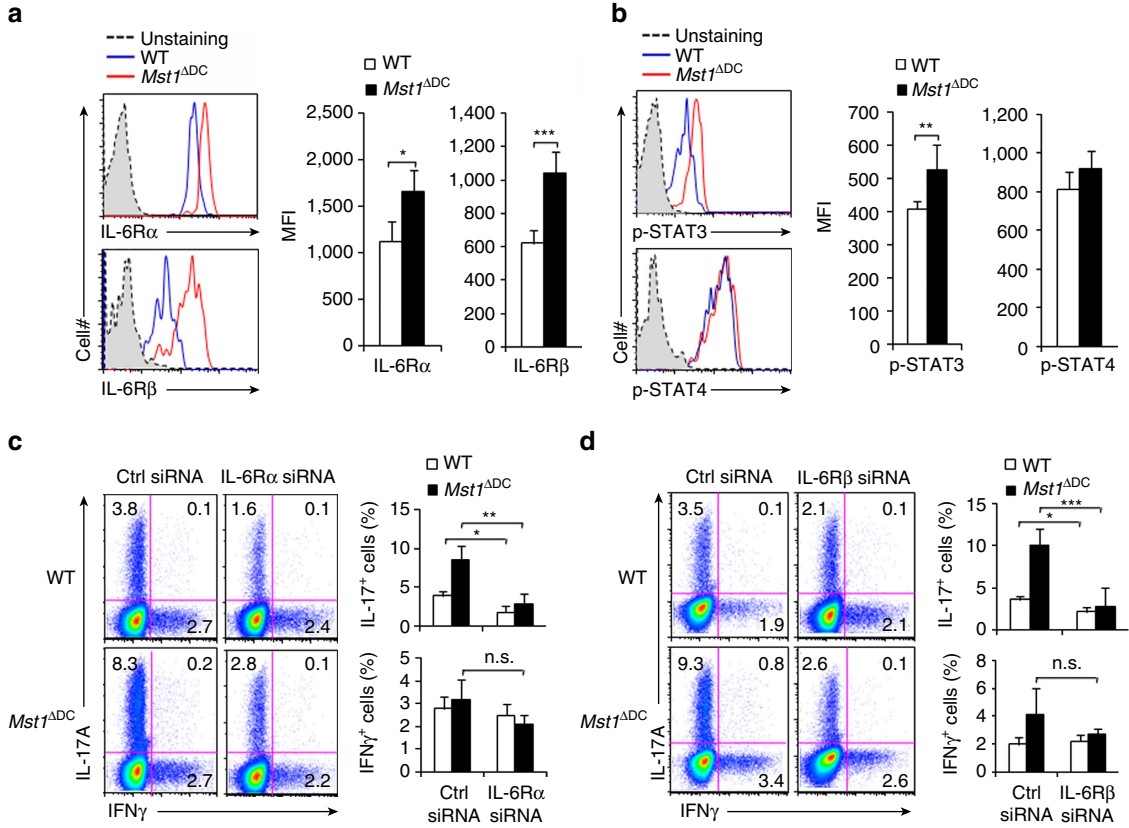

**Figure 7 | T-cell IL-6R-p-STAT3 is responsible for DC MST1 signalling.** (**a**) Expression of IL-6Rα and IL-6Rβ (gp130) in T cells co-cultured with WT or Mst1ΔDC splenic DCs for 3 days. Right, the mean fluorescent intensity (MFI) is summarized. (**b**) Intercellular staining of p-STAT3 and p-STAT4 in T cells co-cultured with WT or Mst1ΔDC splenic DCs for 3 days. Right, MFI of the indicated protein is summarized. (**c,d**) Sorted CD4+ T cells were transfected with control or IL-6Rα (**c**) and IL-6Rβ (**d**) siRNA vector and stimulated with WT or Mst1ΔDC splenic DCs for 5 days. The intercellular staining of IL-17 and IFNγ in T cells performed with FCM and a typical figure shown. Right, the proportion of IL-17 and IFNγ in T cells. Data are representative of three to four independent experiments (mean ± s.d.; n = 4). *P < 0.05, **P < 0.01 and ***P < 0.001, compared with the indicated groups. n.s., not significant. P-values were determined using Student's t-tests.

Th17-dependent autoimmunity. We believe that the phenotypic discrepancy of the MST1 germ line and DC-specific deficiency are likely due to one of the following reasons. One possibility is that the germ line knockout of MST1 may lead to an adaptive and compensatory rewiring of downstream signalling pathways during early development, likely before the development of the DC lineage. On the other hand, such adaptive and compensatory responses are less likely to occur when MST1 is deleted in mature and terminally differentiated DC (Cd11c-Cre plays the major role at this stage). An alternative possibility is that MST1 deficiency in other immune compartments, likely in T cells, may result in an opposite T-cell response and counteract MST1 deficiency in DCs that elicits the T-cell response. A detailed analysis of DCs and other immune compartments in MST1 germ line knockout will be required to test these ideas.

In summary, it is shown that a targeting of MST1 in DCs promotes IL-6 along with the expression of IL-6Rα/β and STAT3, thereby contributing to the programming Th17 cell differentiation in the inflammation that arises in both autoimmunity and fungal infection. Thus, our results define the essential nature of the MST1–p38MAPK–MK2/MSK1–CREB pathway in DCs for inducing Th17 cell differentiation, with implications for targeting DCs as an approach to treatment of disorders of the immune system and immune-associated diseases.

## Methods
**Mice.** All animal experiments were performed with the approval of the Animal Ethics Committees of Beijing Normal University and Fudan University. 2D2

TCR-transgenic mice, IL-6−/− and p38+/− mice were obtained from the Jackson Laboratory (Bar Harbor, ME, USA). OT-II TCR-transgenic mice were obtained from Center of Model Animal Research at Nanjing University (Nanjing, China). CD45.1 and Thy1.1 mice were obtained from the Beijing University Experimental Animal Center (Beijing, China). C57BL/6 Mst1flox/flox and Cd11c-Cre mice were described previously[18,32]. C57BL/6 mice were obtained from the Beijing Weitonglihua Experimental Animal Center (Beijing, China) and Fudan University Experimental Animal Center (Shanghai, China). All mice had been backcrossed to the C57BL/6 background for at least eight generations and were used at an age of 6–12 week, age and sex matched male or female mice unless otherwise noted in the figure legend. WT control mice were of the same genetic background and, where relevant, included Cre+ mice to account for the effects of Cre (no adverse effects due to Cre expression itself were observed in vitro or in vivo).

**Mouse EAE induction and CNS lymphocyte isolation.** Mice were immunized s.c. with 100 μl of emulsified IFA supplemented with 500 μg Mycobacterium tuberculosis H37Ra (DIFCO) and 100 μg of MOG35–55, and received an i.p. injection of 200 ng pertussis toxin (List Biological Laboratories) at the time of immunization as well as 48 h later. The mice were observed daily for clinical signs and scored[38]. Mice were scored for clinical signs of disease on a daily basis using the following scores: 0 = normal behaviour, 1 = distal limp tail, 1.5 = complete limp tail, 2 = disturbed righting reflex, 3 = ataxia, 4 = early paralysis (hind legs), 5 = full paralysis and 6 = moribund or death. For the ex vivo recall response, mice were immunized as above and 7–8 days later, draining LN cells were stimulated with the cognate peptide for 2–3 days for proliferation, RNA and cytokine secretion assays, or were labelled with CFSE (Invitrogen, Carlsbad, CA, USA) and stimulated with antigen for 5 days followed by intracellular staining. T-cell proliferation was determined by pulsing the cells with [3H] thymidine at a dose of 1 Ci per well in a 96-well plate for the final 12–16 h of culture. Cytokine secretion was determined using Elisa (R&D system).

CNS leukocyte isolation was performed[40]. Mice were perfused with 25 ml 2 mM EDTA in PBS to remove blood from internal organs. The spinal columns were dissected, cut open and the intact spinal cord separated carefully from the

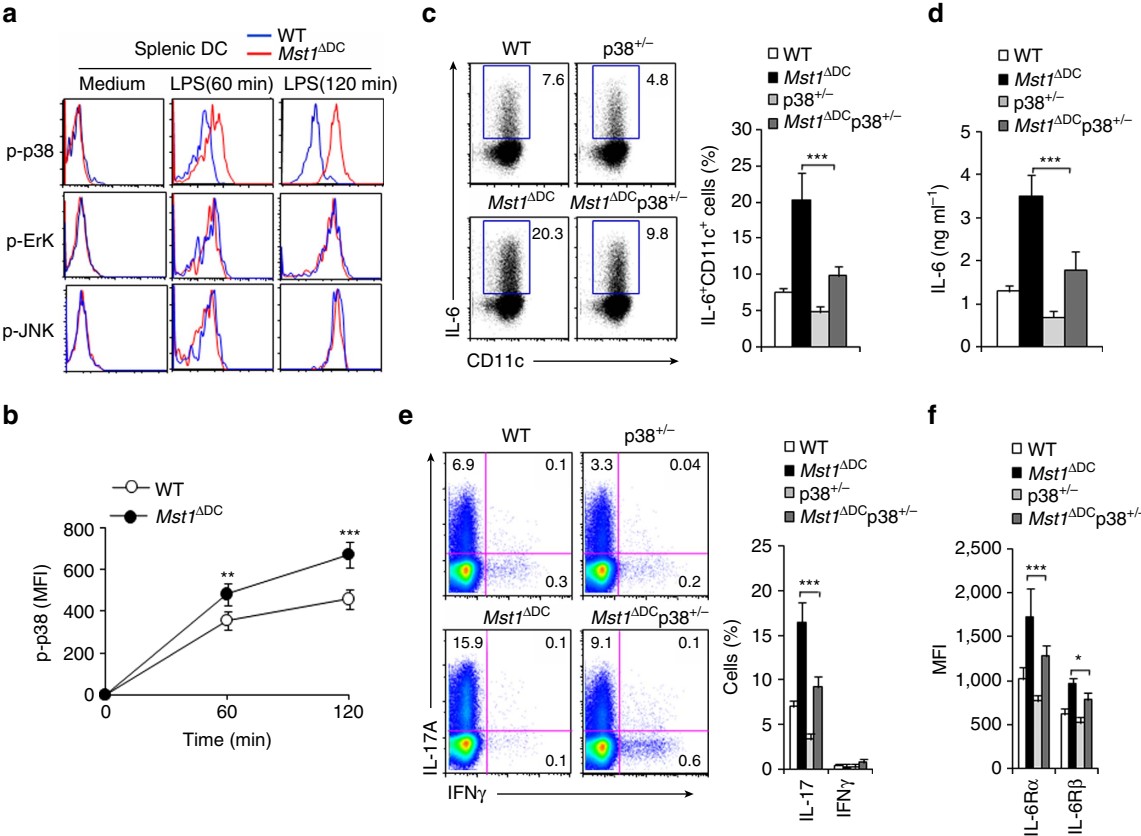

**Figure 8 | MST1 regulates IL-6 production through p38MAPK.** (**a**) Intercellular staining of the indicated proteins in splenic DCs following LPS stimulation *in vitro*. Expression of the phosphorylation of p38MAPK (MFI) among CD11c$^+$ cells is summarized in **b**. (**c,d**) Intracellular staining of IL-6 in DC isolated from the indicated mice 4 h after the i.p. injection of 10 mg kg$^{-1}$ LPS (**c**). Right, the proportion of IL-6$^+$ cells in DCs is summarized. The serum IL-6 level in the indicated mice (**d**). (**e**) Intracellular staining of IL-17 and IFNγ in T cells co-cultured with splenic DCs isolated from indicated mice was determined with FCM. Right, proportion of IL-17 and IFNγ in CD4$^+$ T cells is summarized. (**f**) Expression of the indicated gene in T cells was determined with FCM and the MFI summarized. Data are representative of four independent experiments (mean ± s.d.; n = 3–5). *P < 0.05, **P < 0.01 and ***P < 0.001, compared with the indicated groups. P-values were determined using Student's t-tests.

vertebrae. The spinal cord was cut into small pieces and placed in 2 ml digestion solution containing 10 mg ml$^{-1}$ Collagenase D (Roche) in HBSS. Digestion was performed for 45 min at 37 °C with brief vortexing every 15 min. At the end of digestion, the solution was mixed thoroughly and passed through a 40 μm cell strainer. The cells were washed once in PBS, placed in 6 ml of 38% Percoll solution and pelleted for 20 min at 2,000 r.p.m. Pellets were resuspended in buffer or medium for subsequent analysis.

**Mouse fungal infection.** For fungal infection, mice were injected intravenously with 2 × 10$^5$ live *C. albicans* yeast. After 9 days, mice were killed for analysis. Infected kidney samples were fixed in 4% paraformaldehyde, embedded in paraffin and stained with H&E. Splenocytes were harvested and stimulated with 10$^6$ per ml heat-killed *C. albicans* for 48 h for intracellular staining or RNA isolation.

**Cell isolation from gut-associated lymphatic tissues.** The isolation of lamina propia (LP) lymphocytes was performed[16]. The small intestine (SI) and large intestine (LI) were removed, opened longitudinally and cut into pieces. After vigorous shaking in HBSS containing EDTA, the supernatant containing epithelial cells and IELs was discarded. The remaining intestinal pieces were digested with Collagenase D (Worthington) and pelleted. The pellet was resuspended and placed in a Percoll gradient, and after centrifugation, the interface containing the LP lymphocytes was collected for further analysis.

**Cell adoptive transfer.** Naive T cells 2 × 10$^6$ (CD4$^+$CD62L$^{high}$CD44$^{low}$CD25$^-$) from OT-II or 2D2 TCR-transgenic mice were sorted and transferred into *Mst1*$^{flox/flox}$*Cd11c-Cre*$^-$ (WT) and *Mst1*$^{flox/flox}$*Cd11c-Cre*$^+$ (*Mst1*$^{ΔDC}$) mice. After 24 h, recipient mice were injected s.c. with OVA$_{323-339}$ or MOG$_{35-55}$ in the presence of Complete Freund's adjuvant (CFA; Difco), LPS (Sigma). On day 8–9 after immunization, draining LN cells were harvested and stimulated with the cognate peptide for 2–3 days for cytokine mRNA and secretion analyses, or pulsed with PMA and ionomycin for 5 h for intracellular staining donor-derived T cells.

**Cell cultures and flow cytometry.** Spleens were digested with Collagenase D and DCs (CD11c$^+$TCR$^-$CD19$^-$NK1.1$^-$F4/80$^-$Ly6G$^-$) were sorted on a FACS caliber (Becton Dickinson, CA, USA). Lymphocytes were sorted to enrich for naive T cells. For DC-T-cell co-cultures, DCs and T cells (1:10) were mixed in the presence of 1 μg ml$^{-1}$ OVA$_{323-339}$ peptide and 100 ng ml$^{-1}$ LPS. After 5 days of culture, live T cells were stimulated with PMA and ionomycin for intracellular cytokine staining, or with plate-bound α-CD3 to measure cytokine secretion and mRNA expression. T-cell proliferation was determined by pulsing the cells with $^3$H-thymidine for the final 12–16 h of culture. For drug treatments, cells were incubated with vehicle, SB205830 (5 μM; Calbiochem), U0126 (10 μM; Calbiochem) for 0.5–1 h before stimulation. For cytokine treatment, cultures were supplemented with 20 ng ml$^{-1}$ IL-6 (R&D system). Flow cytometry was performed with antibodies from eBioscience or BD Biosciences. Anti-CD11c APC (clone N418; 1:400), anti-CD11c PE (clone N418; 1:400), anti-CD11c FITC (clone N418; 1:100), anti-CD4 APC-Cy7 (clone GK1.5; 1:100), anti-CD8α FITC (clone 53-6.7; 1:200), anti-CD11b FITC (clone M1/70; 1:200), anti-Ly6G PE (clone RB6-8C5; 1:500), anti-F4/80 PE (clone BM8; 1:400), anti-CD19 PE (clone 1D3; 1:400), anti-TCR FITC (clone H57-597; 1:200), anti-CD44 FITC (clone IM7; 1:200), anti-CD62L APC (clone MEL14; 1:400 ), anti-CD80 APC (clone 16-10A1; 1:400), anti-CD86 APC (clone GL1; 1:400), anti-CD40 APC (clone 1C10; 1:400), anti-CD54 FITC (clone YN1/1/7.4; 1:200 ), anti-MHCII APC (clone AF6-120.1; 1:300), anti-CD45 APC (clone 30-F11; 1:400), anti-NK1.1 PE (PK136; 1:400), anti-IL-17A APC (clone eBio17B7; 1:200), anti-IFNγ PE (clone XMG1.2; 1:200), anti-IL-4 PE (clone 11B11; 1:200), anti-Foxp3 PE (clone FJK-16s; 1:50). The anti-IL-6Rα PE (clone MP5-20F3; 1:100) was from Biolegend and the anti-IL-6Rβ APC (clone MAB4681; 1:100) from R&D systems. All antibodies were diluted in 2% bovine serum albumin. Flow cytometry data were acquired on a FACSCalibur (Becton Dickinson) or an Epics XL bench-top flow cytometer (Beckman Coulter, CA, USA) and the data analysed with FlowJo (Tree Star, San Carlos, CA, USA).

**RNA analysis.** RNA was extracted with an Rneasy kit (QIAGEN, Dusseldorf, Germany) and the cDNA synthesized using SuperScrip III reverse transcriptase

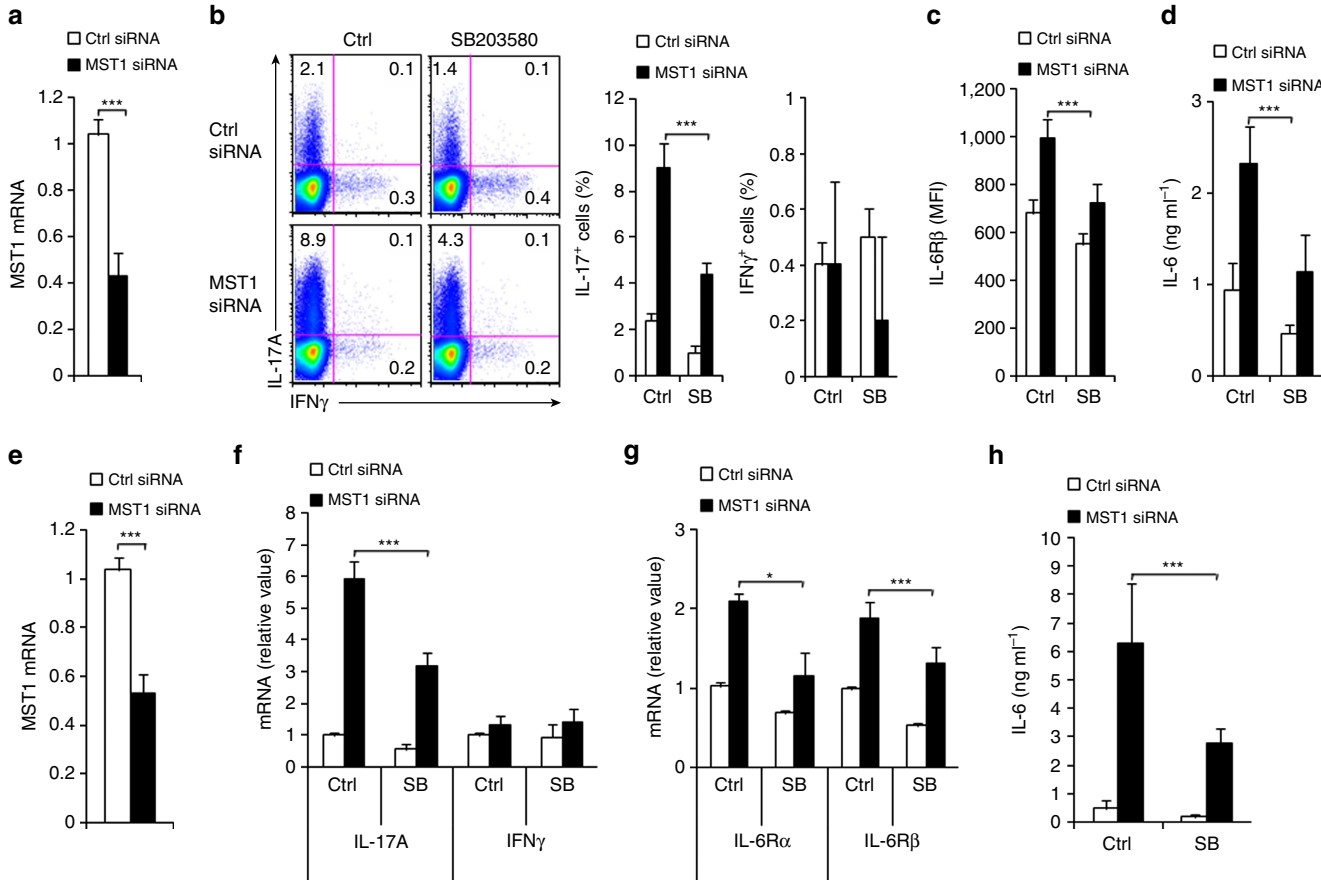

**Figure 9 | Mouse and human DC MST1 modulates T-cell differentiation.** Mouse (**a**) or human (**e**) DCs transfected with MST1-specfic or control siRNA and mRNA of indicated gene were determined with qPCR. Levels in the control groups were set to 1. Mouse DCs (**b–d**) or human DCs (**f–h**) pulsed with LPS (100 ng ml$^{-1}$) were co-cultured with mouse or human T cells, respectively. The expression of IL-17 and IFNγ in T cells was determined with FCM (**b**) or qPCR (**f**); MFI of the indicated gene in T cells (**c**) and mRNA of the indicated genes in T cells (**g**); the levels of IL-6 in the indicated supernatant (**d,h**) were determined. Data are representative of three independent experiments (mean ± s.d.; $n = 4$-6). *$P < 0.05$ and ***$P < 0.001$, compared with the indicated groups. $P$-values were determined using Student's $t$-tests.

(Invitrogen). An ABI 7900 real-time PCR system was used for quantitative PCR (qPCR), with the primer and probe sets obtained from Applied Biosystems (Carlsbad, CA, USA).The primers used in the present study are listed in Supplementary Table 1. Results were analysed using SDS 2.1 software (Applied Biosystems). The cycling threshold value of the endogenous control gene (*Hprt*1, encoding hypoxanthine guanine phosphoribosyl transferase) was subtracted from the cycling threshold. The expression of each target gene is presented as the fold change relative to that of control samples[68].

**Protein analysis.** For detection of phosphorylated signalling proteins, purified cells activated with LPS (Sigma) were immediately fixed with Phosflow Perm buffer (BD Biosciences) and stained with PE or APC directly conjugated with an antibody to Erk phosphorylated at Thr202 and Tyr204 (clone 20A; BD Biosciences), p38MAPK phosphorylated at Thr180 and Thr182 (clone D3F9), JNK phosphorylated at Thr183 and Tyr185 (clone G9), STAT4 phosphorylated at Tyr701 and Ser727 (clone 58D6), STAT3 phosphorylated at Tyr705 (clone D3A7), AKT phosphorylated at Ser473 (clone D9E), AKT phosphorylated at Thr308 (clone D25E6), S6 Ribosomal Protein phosphorylated at Ser235/236 (clone D57.2.2E; all from Cell Signaling Technology and diluted at 1:100 in 2% bovine serum albumin), as described previously[16]. Immunoblot analysis was performed as described[68] The primary antibodies against MST1 (catalogue number 3682), NF-kB p65 (clone D14E12), p38MAPK (catalogue number 9212), p-p38MAPK (Thr180/Thr183; clone 3D7), p-MK2 (Thr334; clone 27B7), p-MSK1 (Thr581; catalogue number 9595), p-CREB (Ser133; clone 87G3), p-C/EBPβ (Thr235; catalogue number 3084), p-c-Fos (Ser32; catalogue number 5348) and p-IκBα (Ser32; clone 14D4; all from Cell Signaling Technology and diluted at 1:1,000 in 2% bovine serum albumin) and along with an anti-β-actin (clone AC-15; Sigma-Aldrich; 1:2,000) antibody. Images have been cropped for presentation and full-size blot is shown in Supplementary Fig. 22.

**Retroviral transduction.** MST1 was cloned into a MSCV retroviral vector (Clontech Laboratories, Mountain View, CA, USA)[17,69]. Phoenic-Eco packaging

cells (ABP-RVC-10001; Allele Biotechnology, San Diego, CA, USA) were transfected with Lipofectamine (Invitrogen) and recombinant retrovirus was collected 48 h after transfection. After sorted splenic DCs were stimulated for 24 h with LPS (Sigma), DC cells were transduced with retroviral supernatant by spin inoculation (650 g for 1 h). Cells were sorted based on the expression of GFP by flow cytometry and cytokine production and gene-expression analysis were performed.

**Gene knockdown with RNAi.** A gene-knockdown lentiviral construct was generated by subcloning gene-specific short RNA (siRNA) sequences into lentiviral siRNA expression plasmids (pMagic4.1)[70]. Lentiviruses were harvested from culture supernatant of 293T cells transfected with the siRNA vector. Sorted splenic DCs or OT-II CD4$^+$ T cells were infected with recombinant lentivirus and GFP-expressing cells were isolated using fluorescence sorting 48 h later. Gene expression was confirmed using qPCR. The sorted DCs and T cells with either the control or siRNA vectors were used for functional assays.

**Human DC and T-cell cultures.** For assays of human DC-mediated T-cell activation and differentiation, normal human DCs (CC-2701; Lonza) were cultured and their populations expanded for 5 days with human granulocyte-macrophage colony-stimulating factor (GM-CSF) and IL-4 (R&D system), followed by stimulation for 24 h with LPS (Sigma). DCs were washed extensively and cultured with human cord blood CD4$^+$ T cells (2C-200; Lonza) at a ratio of 1:10. After 7 days of culture, live T cells were purified and then stimulated either with PMA and ionomycin for intracellular cytokine staining for 5 h or with plate-bound anti-CD3 for analysis of mRNA expression.

**Statistical analysis.** All data are presented as the mean ± s.d. Student's unpaired $t$-test was applied for comparison of means to identify differences between groups. Comparison of the survival curves was performed using the log-rank (Mantel–Cox) test. A $P$-value (alpha-value) of < 0.05 was considered to be statistically significant.

**Data availability.** The authors declare that the data supporting the findings of this study are available within the article and its Supplementary Information files.

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

## Acknowledgements

Our research is supported by grants from the National Natural Science Foundation for General Programs of China (31671524, G.L.); National Basic Research Program of China (2013CB945300, W.T.); National Natural Science Foundation for General Programs of China (31171407 and 81273201, G.L.), Key Basic Research Project of the Science and Technology Commission of Shanghai Municipality (12JC1400900, G.L.), and Innovation Program of Shanghai Municipal Education Commission (14ZZ009, G.L.).

## Author contributions

C.L., Y.B. and Y.L., designed and conducted the experiment with cells and mice, analysed data; H.Y., Q.Y., J.W. and Y.W. conducted the experiments with mice, analysed data; H.S., A.J., Y.H., L.H., J.Z. and S.L. assisted with analysis and interpretation of experiments and results; W.T. contributed to providing the *Mst1*^flox/flox mice and revised the manuscript and G.L. developed the concept, designed and conducted the experiments with cells and mice, analysed data, wrote the manuscript and provided overall direction.

## Additional information

**Competing financial interests:** The authors declare no competing financial interests.

