## [Peer Review File · Nature Communications]

Reviewers' comments:

Reviewer #1, expert in innate instruction of adaptive immunity (Remarks to the Author):

Review of Li et al.

In this study, the authors analyze the role of MST1 in dendritic cells (DCs) in the regulation of CD4 T cell responses. Mice containing a DC-specific deletion of MST1 generate enhanced Th17 responses in several settings. The authors identify increased expression of IL-6 in MST1-deficient DCs as the driver of this enhancement and link the increase of IL-6 to a better phosphorylation of p38. This study therefore identifies MST1 in DCs as a negative regulator that controls the generation of Th17 responses. In recent years, MST1 has received considerable attention. Despite this interest, the role of MST1 in bridging innate and adaptive immunity has remained poorly understood. The current study is timely as it explores this aspect *in vivo*, using an elegant mouse model. In addition, the study dissects the role of MST1 in this process in a thorough manner. Based on these strengths, it should be considered for publication in *Nature Communications*. However, the study also shows some weaknesses that should be addressed first:

Main points:

A) Many of the experiments are superficially described and the experimental conditions are often unclear. Moreover, the language is poor. These craftsmanship issues make it at times difficult to evaluate the experiments and results properly.

B) Along the same lines, the authors present the adoptive T cell experiments (page 11) but neither mention the number of transferred T cells in the main body of the text nor in the figure legend or material and methods. The transfer of a high number of transgenic T cells is not physiological and can lead to experimental artefacts, which is particularly true for innate signaling molecules such as IL-6. This issue could be easily addressed by measuring the endogenous CD4 T cell response following MOG or OVA/CFA immunization and subsequent restimulation *ex vivo* with protein in the presence of irradiated splenocytes and subsequent ELISA (this antigen-specific approach would also control for possible by-stander effects that can occur upon re-stimulation with PMA/ionomycin).

C) The authors propose increased production of IL-6 by MST1-deficient DCs as the mechanism for the enhanced Th17 response. The presented data are indeed suggestive of this interpretation. However, this interpretation fails to account for the possibility that MST1 also controls other aspects of the DC response. The manipulation of IL-6 production by DCs or IL-6 signaling in T cells (Fig. 6-7) is somewhat unsatisfying because it automatically affects Th17 differentiation since IL-6 is required for Th17 differentiation itself. This caveat is underscored by the observation that even in IL-6-deficient DCs, MST1 has an effect on Th17 differentiation (Fig. 6D). Moreover, the authors propose an IL-6-dependent feed-forward loop that leads to increased IL-6R expression on CD4 T cells and phosphorylation of STAT3 (Fig. S11-12). If MST1 indeed controls Th17 differentiation by using IL-6 as a dial, then IL-6R downregulation in T cells should also affect the Th17 levels in co-cultures with WT DCs. It does not. In light of these observations, the authors should attempt to strengthen the rationale for IL-6 as the major MST1-dependent control mechanism:

1) The authors only measure the production of the classic Th17 cytokines IL-6, IL-1, IL-23, and TGF β , of which only IL-6 expression is enhanced in the absence of MST1. What about other cytokines or factors? What about other aspects of DC biology, e. g. apoptosis?

2) Why do WT DCs induce pSTAT3 as effectively in CD4 T cells with reduced IL-6R expression as in WT CD4 T cells (Fig. S11-12)? Is this an effect that depends on the dose of IL-6? Perhaps a titration of IL-6 may help here.

3) IL-6 has also been implicated in the activity of Tregs as well as the generation of Th1 responses. If DC-specific MST1 exerts its influence on CD4 T cells mainly through IL-6, one should also see an effect on Th1 responses. The authors measured IFN γ production but did not observe any differences. However, they measured IFN γ production always under Th17 inducing conditions (both in vivo and in vitro). What about bona fide Th1 responses in vivo?

Reviewer #2, expert in Hippo pathway (Remarks to the Author):

In this manuscript, Li et al. reported that MST1 deficient DC produced higher amount of IL-6, probably by an increased phosphorylation of p38MAPK, and upregulated IL-6R α/β expression and phosphorylation of STAT3 in responding CD4+T cells and contributed to induce TH17 differentiation. MST1 DC conditional knockout mice died earlier and developed spontaneous autoimmune diseases which might due to increased Th17 cell responses. Enhanced IL-6 production in MST1 deficient DC cells is the key finding in this paper. However, this finding cannot be considered novel as it has been previous reported Hippo signaling could negatively regulate IL-6 production in several different cell types; and furthermore the authors did not provide any more convincing mechanism studies to address how MST1 negatively regulates IL-6 production in DC cells. In general, the study provides a lot of correlations and phenomenology, but little definitive evidence to support the conclusion.

Major Concerns:

(1) Previous studies showed that MST1 global knockout displayed relative normal survival rate as compared to that of WT control. It is unclear why MST1 Δ DC conditional knock out mice had earlier mortality rate (Fig.1A). The authors might need exclude the possibility of other genes disrupted by CD11c-Cre transgene which might play an import role in increased mortality rate upon the removal of MST1. Moreover, the authors should check if CD11c-MST1 transgene could rescue the phenotype.

(2) Fig. S2, besides CD11c+ cells, whether knockout of MST1 in DC affects the compositions of other immune cells (i. e. Macrophages, neutrophils, T cells, B cells et al) in MST1 Δ DC conditional knock out mice?

(3) In mice, MST1 is an important regulator of the adhesion and migration of T cells. The authors only checked the CD11c+ cell in spleen. What about other tissues, such as intestine or spinal cord?

(4) Whether MST1 affects the expression levels of co-stimulatory molecule expressions (CD80, CD86, CD40 or CD54) in MST1 deficient DC upon LPS stimulation or upon antigen processing?

(5) It is too casual to claim that MST1 is not involved in regulating antigen presentation of DCs. As in Fig 4, antigen MOG or OVA specific T cells were generated in WT or MST1 Δ DC mice in which many other WT cells could play as antigen presenting cells. In vitro antigen loading and DC-T cells co-culture experiments need to be done for determining the antigen presentation ability of MST1 deficient DC.

(6) The authors claimed that p38 activation was mainly responsible for increased IL-6 production when MST1 was deleted. How about NF- κ B pathway and other regulators of IL-6 induction?

Point by point response to the referees' comments (our response are in blue)

Reviewer #1, expert in innate instruction of adaptive immunity (Remarks to the Author):

Review of Li et al.

In this study, the authors analyze the role of MST1 in dendritic cells (DCs) in the regulation of CD4 T cell responses. Mice containing a DC-specific deletion of MST1 generate enhanced Th17 responses in several settings. The authors identify increased expression of IL-6 in MST1-deficient DCs as the driver of this enhancement and link the increase of IL-6 to a better phosphorylation of p38. This study therefore identifies MST1 in DCs as a negative regulator that controls the generation of Th17 responses. In recent years, MST1 has received considerable attention. Despite this interest, the role of MST1 in bridging innate and adaptive immunity has remained poorly understood. The current study is timely as it explores this aspect in vivo, using an elegant mouse model. In addition, the study dissects the role of MST1 in this process in a thorough manner. Based on these strengths, it should be considered for publication in Nature Communications. However, the study also shows some weaknesses that should be addressed first:

We appreciate the reviewer's recognition of the significance of our work and thoughtful comments. We thank the reviewer for the time and effort involved in assessing our work. Following the reviewer's suggestions and comments, we have added the additional experiments and revised the manuscript.

Main points:

A) Many of the experiments are superficially described and the experimental conditions are often unclear. Moreover, the language is poor. These craftsmanship issues make it at time difficult to evaluate the experiments and results properly.

We thank the reviewer for this suggestion. According to reviewer's suggestions, we have added the detailed experimental descriptions and revised the English text with the help of an English-editing company.

B) Along the same lines, the authors present the adoptive T cell experiments (page 11) but neither mentions the number of transferred T cells in the main body of the

text nor in the figure legend or material and methods. The transfer of a high number of transgenic T cells is not physiological and can lead to experimental artefacts, which is particularly true for innate signaling molecules such as IL-6. This issue could be easily addressed by measuring the endogenous CD4 T cell response following MOG or OVA/CFA immunization and subsequent restimulation ex vivo with protein in the presence of irradiated splenocytes and subsequent ELISA (this antigen-specific approach would also control for possible by-stander effects that can occur upon re-stimulation with PMA/ionomycin).

Thanks for the reviewer's suggestions and comments. Following the reviewer's suggestions, we have added the additional experiments in Fig. 4A and measured the endogenous CD4⁺T cell responses caused by MST1 DC deficient. It showed that MST1 deficient DC enhances T_H17 cell responses, but not T_H1 cell responses.

C) The authors propose increased production of IL-6 by MST1-deficient DCs as the mechanism for the enhanced Th17 response. The presented data are indeed suggestive of this interpretation. However, this interpretation fails to account for the possibility that MST1 also controls other aspects of the DC response. The manipulation of IL-6 production by DCs or IL-6 signaling in T cells (Fig. 6-7) is somewhat unsatisfying because it automatically affects Th17 differentiation since IL-6 is required for Th17 differentiation itself. This caveat is underscored by the observation that even in IL-6-deficient DCs, MST1 has an effect on Th17 differentiation (Fig. 6D). Moreover, the authors propose an IL-6-dependent feed-forward loop that leads to increased IL-6R expression on CD4 T cells and phosphorylation of STAT3 (Fig. S11-12). If MST1 indeed controls Th17 differentiation by using IL-6 as a dial, then IL-6R downregulation in T cells should also affect the Th17 levels in co-cultures with WT DCs. It does not. In light of these observations, the authors should attempt to strengthen the rationale for IL-6 as the major MST1-dependent control mechanism:

We apologize if our previous graph on comparing IL17 percentage in IL-6KO group and MST1^{ΔDC}; IL-6KO double knockout group (previous Fig. 6D, but now is Fig. 6C) was misleading. Now, we have included statistical value on each group. Our results showed that the IL-17 positive percentage is comparable between IL-6KO group and MST1^{ΔDC}; IL-6KO group. These data suggests in IL-6 deficient DCs, MST1 almost has not effects on T_H17 differentiation and indicates IL-6 is largely responsible for the T_H17 induction by MST1 deficient DCs.

We also thank the reviewer for the important suggestions and comments on signaling mechanism downstream of IL-6. Following the reviewer's suggestions, we have added the additional analysis and experiments in the revised manuscript. Our new data showed that the decreased expressions of p-STAT3 in CD4⁺T cells with reduced IL-6R expression, which is related with IL-6 concentration (Fig. S16B and D). Further statistical analysis strengthened our conclusions that less T_H17 induction correlates with reduced IL-6R expression (Fig. 7C-D; *, $P < 0.05$).

1) The authors only measure the production of the classic Th17 cytokines IL-6, IL-1, IL-23, and TGF β , of which only IL-6 expression is enhanced in the absence of MST1. What about other cytokines or factors? What about other aspects of DC biology, e. g. apoptosis?

We appreciate the reviewer's suggestions. According to reviewer's suggestions, we have examined the other cytokine expressions induced by MST1 deficient DC (Fig. 6B) and investigated the apoptosis (Fig. S4A) and proliferation (Fig. S4B) induced by MST1 deficient DC. It showed that MST1 deficient in DC does not affect the apoptosis or proliferation of DCs.

2) Why do WT DCs induce pSTAT3 as effectively in CD4 T cells with reduced IL-6R expression as in WT CD4 T cells (Fig. S11-12)? Is this an effect that depends on the dose of IL-6? Perhaps a titration of IL-6 may help here.

Thanks for the reviewer's important suggestions. We have now included data showing an IL-6 dosage dependent effect on p-STAT3 (Fig. S16B and D).

3) IL-6 has also been implicated in the activity of Tregs as well as the generation of Th1 responses. If DC-specific MST1 exerts its influence on CD4 T cells mainly through IL-6, one should also see an effect on Th1 responses. The authors measured IFN γ production but did not observe any differences. However, they measured IFN γ production always under Th17 inducing conditions (both in vivo and in vitro). What about bona fide Th1 responses in vivo?

Thanks for the reviewer's comments. According to reviewer's suggestions, we have added the additional experiments and showed the IFN γ and Foxp3 expression changes in CD4⁺T cells in MST1 deficient DC old-aged mice (Fig. S5A). DC MST1 deficiency enhanced the percentage of T_H17 and T_H1, but inhibited the percentage of T_{reg} cell in spleen and PLN in the old-aged mice.

Reviewer #2, expert in Hippo pathway (Remarks to the Author):

In this manuscript, Li et al. reported that MST1 deficient DC produced higher amount of IL-6, probably by an increased phosphorylation of p38MAPK, and upregulated IL-6R α / β expression and phosphorylation of STAT3 in responding CD4+T cells and contributed to induce TH17 differentiation. MST1 DC conditional knockout mice died earlier and developed spontaneous autoimmune diseases which might due to increased Th17 cell responses. Enhanced IL-6 production in MST1 deficient DC cells is the key finding in this paper. However, this finding cannot be considered novel as it has been previous reported Hippo signaling could negatively regulate IL-6 production in several different cell types; and furthermore the authors did not provide any more convincing mechanism studies to address how MST1 negatively regulates IL-6 production in DC cells. In general, the study provides a lot of correlations and phenomenology, but little definitive evidence to support the conclusion.

We thank the reviewer for the time and effort involved in assessing our work. We have now included new evidences to further clarify the mechanisms of IL-6 production induced by MST1 deficient. We have determined the activation of MAPK-activated protein kinase 2 (MK2) and mitogen and stress-activated protein kinase 1 (MSK1), two known p38 targets. They showed significantly unregulation of activation in MST1 ^{Δ DC} DC cells compared with WT control cells (Fig. S20A). Additionally, cAMP response-element-binding protein (CREB), activation of transcription factors important for IL-6 expressions, consistently upregulated in MST1 ^{Δ DC} cells, whereas phosphorylation of transcription factor C/EBP β , c-Fos and I κ B α , indicative of transcription factor NF- κ B remains unaltered (Fig. S20A). Consistent with this, modest overexpression of MST1 in DCs resulted in lower and less inactivation of the phosphorylation of MK2, MSK1 and CREB (Fig. S20B). Thus, MST1 is associated with the p38MAPK-MK2/MSK1-CREB signal axis. Importantly, MK2/MSK1/CREB siRNA efficiently silenced their expressions in DC cells and consistently largely reversed the IL-6 production and T_H17 cell differentiation caused by MST1 deficient in DC cells (Fig. S20C-K). Collectively, these data established a p38-MK2/MSK1-CREB signaling axis for the regulation of IL-6 production in DCs for T_H17 differentiation.

Major Concerns:

(1) Previous studies showed that MST1 global knockout displayed relative normal survival rate as compared to that of WT control. It is unclear why MST1 Δ DC conditional knock out mice had earlier mortality rate (Fig.1A). The authors might

need exclude the possibility of other genes disrupted by CD11c-Cre transgene which might play an important role in increased mortality rate upon the removal of MST1. Moreover, the authors should check if CD11c-MST1 transgene could rescue the phenotype.

Thanks for the reviewer's important comments and suggestions. Following the reviewer's suggestions, we set up MST1^{ΔDC}→WT and WT→WT complete chimeras and MST1^{ΔDC}-WT mixed chimeras (1:1) and observed the survival rate of mouse and found that MST1^{ΔDC}→WT complete chimera mice caused similar survival and weight loss as MST1^{ΔDC} mice, but MST1^{ΔDC}-WT mixed chimeras mice significantly recovered these alterations (Fig. S6). These further demonstrating that MST1 deficient in DC plays a critical role in increasing mortality rate.

(2) Fig. S2, besides CD11c+ cells, whether knockout of MST1 in DC affects the compositions of other immune cells (i. e. Macrophages, neutrophils, T cells, B cells et al) in MST1^{ΔDC} conditional knock out mice?

Thanks for the reviewer's suggestions. According to reviewer's suggestions, we have added the additional experiments and observed the compositions of other immune cells (including macrophages, neutrophils, T cells and B cells) in MST1 DC conditional knockout mice in Fig. S2D-E. These data suggests MST1 deficient in DC does not affect the compositions of other immune cells.

(3) In mice, MST1 is an important regulator of the adhesion and migration of T cells. The authors only checked the CD11c+ cell in spleen. What about other tissues, such as intestine or spinal cord?

Thanks for the reviewer's important suggestions. We have already determined the T cell responses induced by DCs from intestine or spinal cord in Fig. S14 and Fig. S15. It showed that DC MST1 deficiency does not alter the DC and T cell number in the local tissue including spleen (Fig. S2A-E), spinal cord (Fig. S14A), and intestine (Fig. S15A). These data suggests MST1 deficient in DC does not affect the migration of DC cells and T cells in vivo under normal physiological condition. However, MST1 deficiency in DCs from spinal cord and intestine consistently promoted the IL-6 production and contributed for the T_H17 cell differentiation in vitro (Fig. S14-15).

(4) Whether MST1 affects the expression levels of co-stimulatory molecule expressions (CD80, CD86, CD40 or CD54) in MST1 deficient DC upon LPS stimulation or upon antigen processing?

Thanks for the reviewer's comments. We have determined the expression levels of co-stimulatory molecules in MST1 deficient DC in the presence of LPS or antigen (Fig. S3C-D). It showed that MST1 deficient does not affect the co-stimulatory molecule expressions in DCs upon LPS stimulation or upon antigen processing.

(5) It is too casual to claim that MST1 is not involved in regulating antigen presentation of DCs. As in Fig 4, antigen MOG or OVA specific T cells were generated in WT or MST1 Δ DC mice in which many other WT cells could play as antigen presenting cells. In vitro antigen loading and DC-T cells co-culture experiments need to be done for determining the antigen presentation ability of MST1 deficient DC.

Thanks for the reviewer's important suggestion. Following the reviewer's suggestion, we have determined the T cell proliferation caused by MST1 deficient DC in co-culture system with T cell in the presence with antigen (Fig. S10A-B). It showed that DC MST1 deficient does not affect the T cell proliferation abilities in DC-T cell in vitro co-culture experiments in the presence of antigen.

(6) The authors claimed that p38 activation was mainly responsible for increased IL-6 production when MST1 was deleted. How about NF- κ B pathway and other regulators of IL-6 induction?

Thanks for the reviewer's important suggestions. We have added the additional experiments and determined the expressions of NF- κ B and AKT-mTOR signals in revised manuscript (Fig. S17A-B and S20A-B). These data showed that NF- κ B and AKT-mTOR signals are not significantly involved in the regulation of IL-6 production induced by MST1 deficiency in DCs.

To further clarified the p38MAPK role in the IL-6 induction by MST1 deficient in DCs. We have added the new experiments and determined the downstream targets of p38MAPK. We have found that the activation of MAPK-activated protein kinase 2 (MK2) and mitogen and stress-activated protein kinase 1 (MSK1), two known p38 targets are significantly strengthened in MST1 Δ DC DC cells compared with WT control cells (Fig. S20A). Additionally, cAMP response-element-binding protein (CREB), activation of transcription factors important for IL-6 expressions, consistently upregulated in MST1 Δ DC cells, whereas phosphorylation of transcription factor C/EBP β , c-Fos and I κ B α , indicative of transcription factor NF- κ B remains unaltered (Fig. S20A). Consistent with this, modest overexpression of MST1 in DCs

resulted in lower and less inactivation of the phosphorylation of MK2, MSK1 and CREB (Fig. S20B). Thus, MST1 regulates p38MAPK-MK2/MSK1-CREB signal axis. Importantly, MK2/MSK1/CREB siRNA efficiently silenced their expressions in DC cells and consistently largely reversed the IL-6 production and T_H17 cell differentiation caused by MST1 deficient in DC cells (Fig. S20C-K). These data further established a p38-MK2/MSK1-CREB signaling axis for the regulation of IL-6 production in DCs for T_H17 differentiation.

Reviewers' comments:

Reviewer #1 (Remarks to the Author):

In the revised version of the manuscript by Li et al., the authors have adequately addressed the majority of the referees' comments. The improved study provides a large body of both in vivo and in vitro work that dissects the cellular and molecular functions of MST1 in DCs. The conclusions of the study are now well supported by the experimental data. While the individual components of the study may not be all that novel, the updated version of the study is nonetheless of considerable interest because it sheds light on the DC-specific role of MST1 in the regulation of IL-6 production in vivo. It supports the notion that IL-6 production by DCs functions as a rheostat for the generation of Th17 immunity and provides insights into an important regulatory circuit that controls IL-6 during DC-T-cell interactions. The study is therefore also of great significance for people with a general interest in the innate control of CD4 T cell responses. Based on these considerations, I support the publication of the study with modest modifications in the text:

1. The finding that DC-specific MST1 KO mice die earlier than WT controls whereas the survival of conventional MST1 KO mice is similar to WT controls is curious and unusual. The authors can replicate this phenotype in bone marrow chimeras but do not provide an explanation for their observation. While a detailed analysis of this observation is probably beyond the scope of the present study, the authors do not talk much about this aspect of their study. I would welcome a dedicated section in the discussion that provides some context for this observation.

2. I still think that there are some craftsmanship and/or grammar issues throughout the text. For example, the sentence on page 21, line 410 lacks a "poorly". It should read as "However, the interplay between adaptive immune cells and innate immune cells remains [poorly] understood."

Reviewer #2 (Remarks to the Author):

The authors have addressed most of the concerns. I will recommend the paper to publish in the journal of Nature communications if they could have additional experiments to solid the conclusion that MST1 in DCs as a negative regulator that controls the generation of Th17 responses. All the following experiments are important as well as easy to do.

1. The authors should look into the active status of T cells including IL-2, CD69 and CD25 in DC Mst1-deficient mice. In vitro co-culture of DC and T cells to see if Mst1-deficient DC cells could trigger T cell activation better than WT DCs.

2. The authors showed the robust induction of IL-6 in Mst1-deficient DCs and then they found IL-6 driven the differentiation of CD4 T cells to Th17 lineages. Since the TH17 and Treg developmental pathways are reciprocally interconnected. The authors should also look into Treg cell subset. The data might not be required to include the paper but better check it.

Point by point response to the referees' comments (our response are in blue)

Reviewer #1 (Remarks to the Author):

In the revised version of the manuscript by Li et al., the authors have adequately addressed the majority of the referees' comments. The improved study provides a large body of both in vivo and in vitro work that dissects the cellular and molecular functions of MST1 in DCs. The conclusions of the study are now well supported by the experimental data. While the individual components of the study may not be all that novel, the updated version of the study is nonetheless of considerable interest because it sheds light on the DC-specific role of MST1 in the regulation of IL-6 production in vivo. It supports the notion that IL-6 production by DCs functions as a rheostat for the generation of Th17 immunity and provides insights into an important regulatory circuit that controls IL-6 during DC-T-cell interactions. The study is therefore also of great significance for people with a general interest in the innate control of CD4 T cell responses. Based on these considerations, I support the publication of the study with modest modifications in the text:

We appreciate the reviewer's recognition of the significance of our work and thoughtful comments. We thank the reviewer for the time and effort involved in assessing our work. Following the reviewer's suggestions and comments, we have added the discussion and revised the manuscript.

1. The finding that DC-specific MST1 KO mice die earlier than WT controls whereas the survival of conventional MST1 KO mice is similar to WT controls is curious and unusual. The authors can replicate this phenotype in bone marrow chimeras but do not provide an explanation for their observation. While a detailed analysis of this observation is probably beyond the scope of the present study, the authors do not talk much about this aspect of their study. I would welcome a dedicated section in the discussion that provides some context for this observation.

We thank the reviewer for this suggestion. According to the reviewer's comments, we have increased the detail discussion about the observation of earlier mouse death in DC-MST1 deficient mice (Highlighted in the text on page 21).

2. I still think that there are some craftsmanship and/or grammar issues throughout the text. For example, the sentence on page 21, line 410 lacks a "poorly". It should read as "However, the interplay between adaptive immune cells and innate immune cells remains [poorly] understood."

We appreciated the reviewer's criticisms. We have corrected the mistakes and re-editing the manuscript with the help of an English-Editing company.

Reviewer #2 (Remarks to the Author):

The authors have addressed most of the concerns. I will recommend the paper to publish in the journal of Nature communications if they could have additional experiments to solid the conclusion that MST1 in DCs as a negative regulator that controls the generation of Th17 responses. All the following experiments are important as well as easy to do.

We thank the reviewer for the time and effort involved in assessing our work. We appreciate the reviewer's thoughtful comments. Following the reviewer's suggestions and comments, we have added the description and revised the manuscript.

1.The authors should look into the active status of T cells including IL-2, CD69 and CD25 in DC Mst1-deficient mice. In vitro co-culture of DC and T cells to see if Mst1-deficient DC cells could trigger T cell activation better than WT DCs.

Thanks for the reviewer's suggestions. We've now included the data and showed that the expressions of IL-2, CD69 and CD25 in T cells co-cultured with MST1-deficient DCs *in vitro* (Fig. S10C). These data suggests that MST1 deficient in DCs doesn't alter the active status of T cells.

2.The authors showed the robust induction of IL-6 in Mst1-deficient DCs and then they found IL-6 driven the differentiation of CD4 T cells to Th17 lineages. Since the TH17 and Treg developmental pathways are reciprocally interconnected. The authors should also look into Treg cell subset. The data might not be required to include the paper but better check it.

We thank and agree with the reviewer's comments on the importance of extending our current findings to Treg cell studies. Respectively, however, as Editor and reviewer suggested that these data go beyond the scope of the current manuscript. We should look into them when we can get opportunities in the future.

REVIEWERS' COMMENTS:

Reviewer #2 (Remarks to the Author):

The authors addressed all of my concerns

Point by point response to the referees' comments (our response are in blue)

Reviewer #2 (Remarks to the Author):

The authors addressed all of my concerns

We thank the reviewer for the time and effort involved in assessing our work.